# CHOrD: Synthesizing Spatially Coherent, House-Scale, Organized, and Diverse 3D Indoor Scenes via Image-Based Layout Guidance

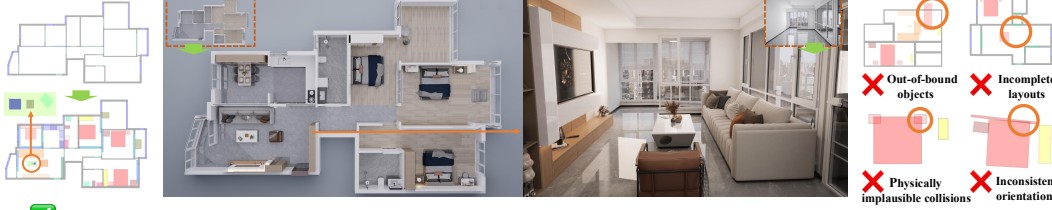

Spatially coherent, house-scale, multi-level 3D indoor scene synthesized with complex floor plan structures and room shapes

Figure 1: CHOrD synthesizes spatially coherent, house-scale, multi-level structured, and diverse indoor scene layouts with complex room shapes and floor plans. Its strong spatial capabilities substantially reduce common artifacts in prior work, such as out-of-bound objects, incomplete layouts, collisions, and inconsistent orientations. CHOrD is fully data-driven, requiring no collision detection, iterative self-correction, or manually crafted rules.

## Abstract

We introduce **CHOrD**, a generative framework for synthesizing spatially coherent, house-scale, hierarchically organized, and diverse 3D indoor scenes. At the core of CHOrD is a two-stage generation paradigm: given a floor plan, CHOrD first synthesizes an intermediate, image-based 2D layout representation, which is subsequently transformed into a graph-based scene structure. In contrast to existing tabular-based or LLM-based generative models, the enhanced spatial capabilities of CHOrD substantially reduce several long-standing artifacts frequently observed in prior work—such as physically implausible collisions, out-of-bound objects, inconsistent orientations, or incomplete layouts missing essential object placements. Furthermore, unlike existing methods, CHOrD can be conditioned on complex, irregular room shapes and is robust in synthesizing house-wide layouts that adhere to both geometric and semantic floor plan structures. We also introduce a novel layout dataset with expanded coverage of object categories and room configurations, as well as significantly improved data quality. CHOrD achieves state-of-the-art performance on both the 3D-FRONT dataset and our proposed dataset, excelling in spatial coherence, quality, and diversity, without relying on collision detection, iterative re-generation for self-correction, or predefined rules.

## 1 Introduction

Generative 3D indoor scene synthesis and virtual indoor digital twin creation (Merrell et al., 2011; Yu et al., 2011; Fisher et al., 2012; Qi et al., 2018; Zhang et al., 2018; Li et al., 2018a; Ritchie et al., 2019; Wang et al., 2019; Yao et al., 2024; Min et al., 2024; Vaswani et al., 2017; Hu et al., 2020; Wang et al., 2020; Paschalidou et al., 2021; Leimer et al., 2022; Tang et al., 2024; Lin & Mu, 2024) play increasingly vital roles not only in creative and technical workflows such as interior design, architectural planning, and virtual and augmented reality, but also in advancing embodied AI by providing scalable simulated environments for training and testing. This approach facilitates rapid prototyping, reduces manual labor, lowers deployment costs, and accelerates iteration. Despite recent advances in neural volumetric representations (Mildenhall et al., 2020; Kerbl et al., 2023), classic mesh-based assets remain the predominant 3D digital twin representation in these domains due to superior rendering quality, direct interactivity, and explicit geometric structures. Consequently, existing pipelines (Zhang et al., 2018; Ritchie et al., 2019; Wang et al., 2019; Paschalidou et al.,

2021; Tang et al., 2024; Lin & Mu, 2024) primarily follow a **procedural generation paradigm**, constructing a *scene graph* or *object list* for the scene layout, with each node containing detailed specifications for individual objects, such as categories, locations, and attributes. These objects can then be retrieved from a CAD asset dataset and rendered or interacted with using various graphics and physics engines to create simulation-ready scenes. Therefore, synthesizing diverse and logical scene layouts has been a **core aspect** of high-quality virtual indoor digital twin creation.

However, a fundamental limitation of existing methods that **directly** construct scene graphs or object lists—either by a tabular generative model (Li et al., 2018a; Ritchie et al., 2019; Wang et al., 2019; Paschalidou et al., 2021; Tang et al., 2024; Lin & Mu, 2024) or by an LLM writing configuration files (Yang et al., 2024b; Feng et al., 2023)—is their **limited capability** in preventing various commonly observed spatial artifacts during the generation process, such as physically implausible collisions, out-of-bound objects, inconsistent orientations, and incomplete layouts missing major object placements, as listed in Figure 1 and demonstrated in Figure 5. While post-processing steps such as collision detection can be performed, they are computationally expensive and require iterative re-generation to correct such artifacts, consuming significant GPU resources and LLM tokens, which severely limits their scalability. Prior work has also attempted to prevent spatial artifacts using manually defined rules (Deitke et al., 2022; Raistrick et al., 2024), but this approach lacks generalizability to arbitrary scenes and cannot be used to learn desirable layout distributions from data. Another critical yet frequently overlooked limitation of existing methods is their **restriction** to simplistic rectangular room shapes or single-room layouts (Zhang et al., 2018; Li et al., 2018a; Ritchie et al., 2019; Wang et al., 2019; Paschalidou et al., 2021; Tang et al., 2024; Lin & Mu, 2024), which fails to account for the complex geometry and overall floor plan structure of a house. Since room shapes, sizes, along with the placements of doorways and windows collectively influence the logical organization of the scene layout, existing approaches neglect key spatial relationships essential for irregular-shaped or multi-room designs. The occurrence of these limitations is not coincidental — current tabular generative models and LLMs have not achieved the **granular spatial understanding** necessary to capture nuanced spatial relationships, such as distinguishing adjacency from intersection, or to faithfully integrate complex floor plan geometries into the generative process.

In this paper, we propose CHOrD, a framework designed to comprehensively enhance **spatial coherence** in synthesizing 3D indoor scene layouts, as highlighted in Figure 1. In particular, CHOrD *i)* substantially reduces various spatial artifacts during the generation process without relying on collision detection, iterative re-generation for self-correction, or pre-defined rules; *ii)* enables house-scale layout generation that adheres to complex geometric and semantic floor plan structures, which are also controllable via multi-modal input; and *iii)* supports a hierarchically structured scene graph representation that seamlessly integrates into existing pipelines. Central to our approach is the synthesis of a *2D image-based* layout representation as an intermediate step in the procedural workflow, which can be subsequently converted into a hierarchical scene graph, **rather than** constructing the graph in a single step. Our key insight is that, compared to the graph representation, which is inherently *tabular*, introducing an intermediate image-based 2D layout representation greatly strengthens the **spatial capabilities** of the generative model. For example, humans can readily spot collisions by examining the top-down view of a layout, whereas simply reviewing a table of bounding box values does not enable such direct assessment. Designers routinely rely on top-down floor plan views to create orderly spatial layouts. In a similar vein, CHOrD effectively recognizes spatial anomalies as *out-of-distribution* (OOD) scenarios, by leveraging powerful image encoders and decoders to internalize spatial priors. This capability enables the empirical elimination of various spatial artifacts during the generation process, as well as the adaptation of complex floor plan structures. We advocate incorporating an intermediate image-based layout representation for all graph-based methods.

We additionally introduce a novel dataset, referred to as the CHOrD dataset, comprising 9,706 scenes with floor plans and scene layouts, approximately 1.4 times larger than 3D-FRONT (Fu et al., 2021a). Compared to 3D-FRONT, the CHOrD dataset expands household item coverage to 26 super-categories from kitchens, bathrooms, and balconies, addressing gaps in 3D-FRONT, which lacks furnishings in these areas and occasionally leaves living rooms or bedrooms unfurnished. It also resolves common issues found in 3D-FRONT, such as misclassified objects, unrealistic placements, and collisions, providing clean layouts without requiring extensive data cleaning.

Finally, CHOrD achieves state-of-the-art performance on both the 3D-FRONT and our proposed datasets, evaluated both qualitatively and quantitatively, particularly in the near-elimination of various spatial artifacts in a single generation step, a prevalent issue in existing methods.

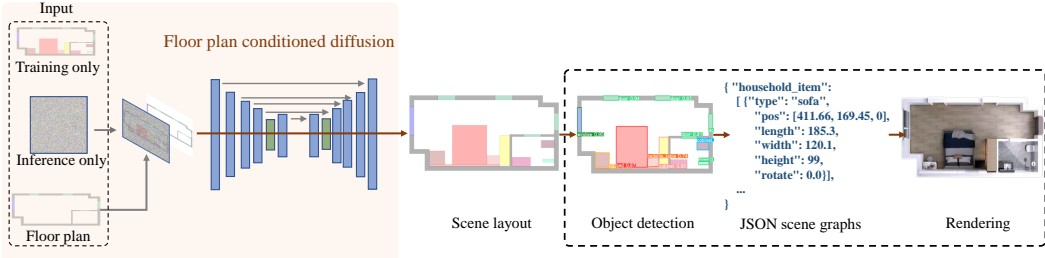

Figure 2: CHOrD starts from a floor plan image as input, where a conditional diffusion model generates a 2D image-based layout. Object detection and segmentation then produce a structured scene graph of spatial relationships and attributes, from which 3D objects are retrieved and rendered into photorealistic, simulation-ready scenes.

## 2 RELATED WORK

**Rule-based methods**  Rule-based methods generate scene layouts via constraint-satisfaction for pre-specified objects (Xu et al., 2002; Deitke et al., 2022) or cost function optimization, manually defined based on interior design principles (Merrell et al., 2011; Yu et al., 2011). While rule-based methods allow for moderate diversity, their non-data-driven nature prevents them from learning complex, unconstrained layout distributions from data.

**Data-driven methods**  With the release of large datasets of 3D indoor scene layouts and associated assets, such as SUNCG (Song et al., 2017), 3D-FRONT (Fu et al., 2021a), SUN3D (Xiao et al., 2013), Matterport3D (Chang et al., 2017), InteriorNet (Li et al., 2018b), Structured3D (Zheng et al., 2020), and 3D-FURNITURE (Fu et al., 2021b), learning-based approaches became dominant, including Bayesian networks and Gaussian mixtures (Fisher et al., 2012), probabilistic grammars (Qi et al., 2018), GANs (Zhang et al., 2018), recursive networks (Li et al., 2018a), CNNs (Wang et al., 2018; Ritchie et al., 2019), graph neural networks (Wang et al., 2019; Yao et al., 2024), transformers (Wang et al., 2020; Paschalidou et al., 2021; Leimer et al., 2022; Para et al., 2023; Sun et al., 2024), and diffusion models (Tang et al., 2024; Lin & Mu, 2024; Maillard et al., 2024; Bokhovkin et al., 2024). Despite different architectures, these models all operate on *tabular representations* of scene layouts (*e.g.*, structured lists or matrices encoding objects and their attributes such as category, position, orientation, and relations) and can thus be categorized as tabular generative methods. Large language models (LLMs) have also been explored for synthesizing layouts (Feng et al., 2023; Yang et al., 2024b), either by sequentially generating the tabular entries of a layout or by producing a configuration file (*e.g.*, a JSON file encoding objects and attributes) as text. Both tabular data and JSON configuration files serve as equivalent representations of a scene graph.

While these methods have achieved compelling results, they remain limited in spatial capabilities, such as not allowing irregular room shapes or complex floor plans, producing object collisions or out-of-bound placements, omitting essential objects, or generating inconsistent object orientations. Although some methods resort to collision detection to address collisions and out-of-bound issues, it does not resolve the other spatial artifacts mentioned above. Moreover, when collisions occur, the typical strategy of these methods is to iteratively re-generate layouts and re-check for collisions until no conflicts remain. If the generative model has a high probability of producing collision artifacts, this iterative process becomes prohibitively time-consuming and computationally expensive. The problem is exacerbated in LLM-based methods, where generation incurs large token costs. Consequently, a generative model for spatially coherent layouts without requiring repeated self-correction is highly desirable and remains an open challenge for data-driven approaches. A comparison of existing methods and our approach is summarized in Table 1. CHOrD employs an intermediate image-based layout representation to guide scene graph generation, excelling at spatial reasoning while remaining compatible with standard graph-based pipelines.

## 3 CHOrD PIPELINE

We propose a novel pipeline for 3D-aware indoor scene synthesis and virtual digital twin creation. As depicted in Figure 2, the pipeline starts with a floor plan description—provided as an 2D im-

| Reference | Layout representation | Method | Predefined rules | Irregular room shapes / house-scale | Iterative collision detection / re-generation for correction |
|---|---|---|---|---|---|
| Xu et al. (2002) | | | Yes | Yes | No |
| Deitke et al. (2022) | Tabular data | Rule-based | Yes | Yes | Yes |
| Merrell et al. (2011); Yu et al. (2011) | | | Yes | No | Yes |
| Fisher et al. (2012) | | Bayesian | No | No | No |
| Qi et al. (2018) | | MRF | No | No | No |
| Zhang et al. (2018) | | GANs | No | No | Yes |
| Wang et al. (2018); Ritchie et al. (2019) | | CNN | No | No | Yes |
| Li et al. (2018a) | Tabular data | RvNN-VAE | No | No | No |
| Wang et al. (2019); Yao et al. (2024) | | VAE | No | No | Yes |
| Wang et al. (2020) | | GNN | No | No | No |
| Tang et al. (2024) Lin & Mu (2024) Yang et al. (2024a) | | Diffusion | No | No | No |
| Wang et al. (2020) Para et al. (2023) Sun et al. (2024) Feng et al. (2025) | | Transformer | No | No | Yes |
| Paschalidou et al. (2021) | | Transformer | No | No | No |
| Yang et al. (2024b) Feng et al. (2023) | Text | LLMs | No | No | Yes |
| **CHOrD (Ours)** | 2D layout image | Diffusion | **No** | **Yes** | **No** |

Table 1: Comparison of layout representations and methods in related work.

age—and uses a conditional diffusion model to generate a corresponding 2D scene layout. The use of this 2D representation enables us to leverage efficient image encoders for layout generation, effectively enhancing spatial coherence and preventing various spatial artifacts. Next, we employ object detectors and segmentation maps to identify individual household items and extract a structured scene graph that hierarchically organizes *multi-level* spatial relationships and object attributes. Finally, the 3D scene objects are retrieved accordingly and rendered to produce photorealistic 3D-consistent images, which can also be deployed in a physics engine for simulation.

## 3.1 DIFFUSION-BASED SCENE LAYOUT GENERATION

We leverage the success of image-based diffusion models (Saharia et al., 2022a; Rombach et al., 2022; Amit et al., 2021) and frame the problem of generating diverse, realistic indoor scene layouts as a conditional image-to-image translation task, as illustrated in Figure 2. Unlike complex scene graphs or tabular formats, natural 2D images serve as a convenient intermediate representation for the layout, easily processed by existing vision tools. Crucially, as 2D images are easy-to-interpret by an appropriate encoder, we can construct a highly effective conditional generative model that accurately captures the data distribution. In 2D images, implausible spatial artifacts are instantly visible and flagged as OOD samples, enabling the model to generate coherent, realistic layouts.

Specifically, given an image of an empty floor plan $\boldsymbol{y}$, we train a diffusion model $\epsilon_\theta(\boldsymbol{x};\boldsymbol{y},t)$ to model the conditional distribution of the corresponding layouts $p(\boldsymbol{x} \mid \boldsymbol{y})$, where $\epsilon_\theta$ is structured as a 2D U-Net, following Ho et al. (2020), with 3 input channels (random noise) and 3 output channels (the predicted layout image). To incorporate floor plan image conditioning, we expand the U-Net input from 3 to 6 channels. During training, a predetermined noise schedule realizes a Markov chain, yielding the diffused sample $\boldsymbol{x_t}(\boldsymbol{x},\boldsymbol{y},t,\epsilon)$, where $\epsilon \sim \mathcal{N}(\boldsymbol{0},\boldsymbol{I})$ and $t \sim \mathcal{U}(0,1)$. The loss function is given by the denoising score matching objective (Ho et al., 2020):

$$\mathbb{E}_{(\boldsymbol{x},\boldsymbol{y})\sim p_{\text{data}},\epsilon\sim\mathcal{N}(\boldsymbol{0},\boldsymbol{I}),t\sim\mathcal{U}(0,1)}\left[\left\|\epsilon_\theta(\boldsymbol{x};\boldsymbol{y},t)-\epsilon\right\|^2\right]. \tag{1}$$

To accelerate the inference process, we adopt DPMSolver (Lu et al., 2022), which enables the generation of high-quality layout results using significantly fewer steps during inference. Additional mathematical background on diffusion models is provided in Appendix A.

## 3.2 HIERARCHICAL SCENE GRAPH EXTRACTION AND OBJECT RETRIEVAL

To generate a scene graph from the candidate layout $\boldsymbol{x} \sim p(\boldsymbol{x} \mid \boldsymbol{y})$, we follow a framework similar to (Lv et al., 2021). As depicted in Figure 3, we start by fine-tuning YOLOv8 (Jocher et al., 2023) to detect the locations and attributes of all objects present in $\boldsymbol{x}$. The color of each object uniquely identifies its category from a set of 28 household item categories and 3 floor plan item categories. Detailed color schemes are listed in Appendix Table 4. The other attributes are populated to produce an object list $\mathcal{O} = (\boldsymbol{o_1}, \boldsymbol{o_2}, \ldots, \boldsymbol{o_n})$, with each node containing object properties such as category, position, orientation, and size. We employ YOLOv8 to simultaneously obtain the segmentation maps for each room type, including living rooms, bedrooms, kitchens, bathrooms, and balconies.

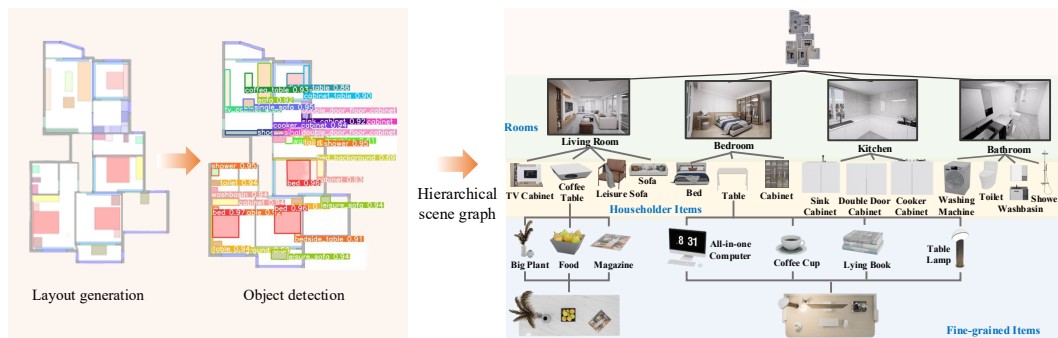

Figure 3: Scene graph extraction and object retrieval.

Given the dimensions and category of each object, we deterministically retrieve an example from a category-specific textured mesh database $\mathcal{D}$[1] such that it has the smallest size difference:

$$\boldsymbol{e_i} = \arg\min_{\boldsymbol{e} \in \mathcal{D}}(\|o_i^x - e_i^x\|^2 + \|o_i^y - e^y\|^2) : o_i^c = e^c, \forall \boldsymbol{o_i} \in \mathcal{O}. \tag{2}$$

The set of retrieved examples $\{\boldsymbol{e_1}, \boldsymbol{e_2}, \ldots, \boldsymbol{e_n}\}$ constitutes the leaf nodes of the scene graph, as shown in Figure 3. To position the objects in each room and construct the hierarchical scene graph, we utilize the semantic detection and segmentation outputs of household items and rooms from YOLOv8. We straighten the edges of the room polygons, similar to (Lv et al., 2021), to reduce uneven lines, and attach doors and windows to these edges, ensuring corrected wall positions that enclose the room.

**Fine-grained layout**  Note that this approach enables CHOrD to generate granular, hierarchical, fine-grained layouts in a multi-level *autoregressive* manner. Specifically, we can apply a separate conditional diffusion model to generate fine-grained layouts, such as placing objects on a coffee table, as illustrated in Figure 3 (bottom right). When generating fine-grained layouts, the conditional input for the diffusion model becomes the top-down views of the upper level (*e.g.*, table boundaries) instead of floor plan images. Additional technical details of fine-grained layout generation are provided in Appendix B.1 and Figure 10.

The advantages of a hierarchical layout structure are threefold. First, this structure allows CHOrD to be seamlessly integrated into widely adopted graph-based pipelines (Tang et al., 2024; Lin & Mu, 2024; Yang et al., 2024a) to enhance spatial coherence, which we strongly advocate. Second, this structure is compatible with a wide range of downstream tasks, such as simulating intricate spatial understanding and navigation (Werby et al., 2024). Finally, this multi-level layout enables CHOrD to also accommodate natural vertical object overlaps, such as placing objects on a coffee table.

**Multi-modal floor planning**  Apart from the main pipeline, the 2D layout of CHOrD enables additional multi-modal controls for the floor plan. Specifically, we provide two types of controls: *text-conditioned* and *open-plan-conditioned* floor planning, which are detailed in Appendix B.1.

**Rendering**  Finally, we convert the structured scene graph into a 3D mesh. The wall and floor materials for each $\boldsymbol{e_i}$ are procedurally sampled, while being aware of the rooms to which they belong. An appropriately sized area light is placed at the center of each room. The UE engine (Epic Games) is subsequently utilized to generate photorealistic renderings. Additional rendering details are provided in Appendix B.2.

The key advantage of the multi-stage pipeline of CHOrD—which synthesizes an image-based 2D layout as an intermediate representation rather than directly generating a scene graph as tabular data —lies in its comprehensively enhanced fine-grained spatial capabilities. CHOrD adapts to complex room shapes and floor plan structures, ensures that the hierarchical spatial relationships between household items are preserved, and reduces common spatial artifacts observed in prior work, such as object overlap, collisions, out-of-bounds placements, incomplete layouts, and orientation inconsistencies (both among objects and between object and room geometry), as validated in Section 5.

---

[1]Note that the selection of this database and its retrieval rules can be flexibly user-specified, enabling custom and advanced functionality.

Figure 4: **Left** - Visualization of three diverse layouts (columns) synthesized by CHOrD for each of the three floor plans (rows). CHOrD is robust to irregular and slanted room shapes. **Right** - Photorealistic rendering of living rooms and bedrooms with identical camera positions and floor plans, highlighting their diversity. The correspondence between the layout on the left and the rendering on the right is indicated by matching colored frames.

# 4  CHORD DATASET

We collected a new large-scale dataset, which we refer to as the CHOrD dataset, of indoor scenes with floor plans and scene layouts, comprising a total of 9,706 design schemes, approximately 1.4 times larger than the 3D-FRONT dataset (Fu et al., 2021a). This dataset was meticulously created by professional interior designers, stored in JSON format, as exemplified in Appendix List 1, including wall lines, doors, windows, and household items covering 26 super-categories across furniture, fixtures, and appliances. In this dataset: **Rooms** are represented as enclosed loops of interior wall lines, defined by 2D coordinates. **Doors, windows, and objects** are represented as 2D bounding boxes, defined by category, 3D coordinates, orientation, and dimensions (length, width, height).

It is important to note that CHOrD dataset is a 3D *layout* dataset rather than a 3D *asset* dataset. The layout primarily focuses on the geometric characteristics (*e.g.*, bounding boxes) and categorical distinctions among objects. While CHOrD dataset is currently linked to a small pool of CAD asset models, users are free to retrieve assets from any large public dataset (3D66, 2013; Fu et al., 2021b) to introduce stylistic variations of objects or simulation-ready URFD files if needed. Similarly, 3D-FRONT has been associated with the 3D-FUTURE dataset Fu et al. (2021b) for this purpose.

CHOrD dataset offers several clear advantages over 3D-FRONT. 3D-FRONT is currently limited to living, dining, and bedroom layouts, lacking kitchen, bathroom, and balcony data. It also contains erroneous layouts such as empty rooms, unnatural object sizes, misclassified items, and unrealistic placements (*e.g.*, objects outside boundaries, lamps on floors, blocked doorways, and object overlaps), requiring extensive data cleaning. In contrast, CHOrD dataset covers all major room types, including fully furnished kitchens, bathrooms, and balconies, and is artifact-free and ready to use. Additional details of the CHOrD dataset are provided in Appendix C, with statistical and visual comparisons to 3D-FRONT in Tables 7–9 and Figures 11–12.

# 5  EXPERIMENTS

| | Dataset | Bedroom | | | | Living Room | | | | Entire House | | | |
|---|---|---|---|---|---|---|---|---|---|---|---|---|---|
| | | FID↓ | KID↓ | POR↓ | PIoU↓ | FID↓ | KID↓ | POR↓ | PIoU↓ | FID↓ | KID↓ | POR↓ | PIoU↓ |
| DiffuScene | 3D-FRONT | 15.91 | 0.04 | 0.1632 | 0.0152 | 45.89 | 0.034 | 0.05 | 0.012 | - | - | - | - |
| InstructScene | 3D-FRONT | 22.35 | 0.02 | 0.2039 | 0.0088 | - | - | - | - | - | - | - | - |
| PhyScene | 3D-FRONT | - | - | - | - | 117.29 | 0.119 | 0.389 | 0.0134 | - | - | - | - |
| CHOrD (ours) | 3D-FRONT | **14.78** | **0.008** | **0.0637** | **0.0008** | **24.15** | **0.018** | **0.0166** | **0.0011** | 12.84 | 0.007 | 0.0106 | 0.0003 |
| DiffuScene | CHOrD dataset | 37.16 | 0.03 | 0.1922 | 0.0038 | 29.97 | 0.02 | 0.0707 | 0.0028 | - | - | - | - |
| InstructScene | CHOrD dataset | 48.59 | 0.05 | 0.3010 | 0.0092 | 46.05 | 0.04 | 0.0908 | 0.0037 | - | - | - | - |
| CHOrD (ours) | CHOrD dataset | **21.86** | **0.02** | **0.1053** | **0.0022** | **26.69** | **0.02** | **0.0185** | **0.0021** | 21.84 | 0.021 | 0.0119 | 0.0005 |

Table 2: Quantitative evaluation of CHOrD against prior approaches, demonstrating superior performance across all metrics and datasets.

We conducted several experiments to assess the performance of CHOrD on layout synthesis in comparison with prior work. We particularly evaluate the effectiveness of CHOrD in nuanced spatial

understanding by assessing its ability to capture collision artifacts as out-of-distribution samples. Next, we demonstrate the versatility of CHOrD in several extended tasks, including fine-grained layout synthesis and multi-model floor planning.

## 5.1 FLOOR PLAN-CONDITIONED SYNTHESIS

**Implementation** We trained CHOrD on four RTX 8000 GPUs with a batch size of 4 for 400 epochs. The initial learning rate was set to 1e-4, with a decay factor of 0.1 every 100 epochs. For the diffusion process, we followed the default configuration of DDPM (Ho et al., 2020), where noise intensity gradually increases from 0 to 1 over 1000 time steps. For the object detection process, we followed the default configuration of YOLOv8 (Jocher et al., 2023). Further implementation details are provided in Appendix D.

**Datasets** We compare CHOrD with baseline methods on both the 3D-FRONT dataset (Fu et al., 2021a) and the proposed CHOrD dataset. The 3D-FRONT dataset consists of 6,813 scenes, of which 4,847 were retained after a cleaning process that excluded layouts lacking furniture, containing objects extending beyond room boundaries, or exhibiting collisions. Prior works (Zhang et al., 2018; Ritchie et al., 2019; Paschalidou et al., 2021; Tang et al., 2024; Lin & Mu, 2024) have applied similar data filtering to remove erroneous scenes from 3D-FRONT. The CHOrD dataset comprises 9,706 scenes and is ready for use without the need for data cleaning or preprocessing. We use 80% of the dataset for training and 20% for testing.

**Baselines** We compare CHOrD with DiffuScene (Tang et al., 2024), InstructScene (Lin & Mu, 2024), and PhyScene (Yang et al., 2024a), all aiming to synthesize 3D indoor scenes with optimized layouts. Note that DiffuScene, InstructScene, and PhyScene are all unable to synthesize house-scale layouts but individual categories of rooms. Additional baselines such as Holodeck (Yang et al., 2024b) and LayoutGPT (Feng et al., 2023) are prompt-based systems built on closed-source pretrained LLMs, making it infeasible to fine-tune them on 3D-FRONT and CHOrD dataset to align with specific object placement distributions for quantitative comparison. Spatial artifacts such as collisions and out-of-bound objects have also been reported in the original papers of Holodeck and LayoutGPT. While other baselines exist (Para et al., 2023; Sun et al., 2024; Maillard et al., 2024; Bokhovkin et al., 2024; Feng et al., 2025), these methods have not released source code.

For evaluation on 3D-FRONT, we used the official pre-trained checkpoints of baseline methods to ensure their optimal performance. Specifically, we used the checkpoint from the DiffuScene unconditional model to generate top-down views of object layouts in bedrooms and living rooms at a resolution of $256 \times 256$, matching the image size generated by our diffusion model. For InstructScene, we similarly used the checkpoint from the unconditional model to generate bedroom views at the same resolution. InstructScene did not release unconditional model checkpoints for living rooms. For PhyScene, we used their checkpoint from the floorplan-conditioned model to generate living room layouts. PhyScene did not release model checkpoints for bedrooms. To ensure fairness in the comparison, the object categories generated by DiffuScene, InstructScene, and PhyScene were remapped to our categorization, as detailed in Appendix Table 6. For evaluation on the CHOrD dataset, we re-trained the unconditional models of DiffuScene and InstructScene on living rooms and bedrooms using their default training configurations. PhyScene did not release its training code.

**Results** We present the qualitative evaluation of all methods in Figure 5, with all results randomly selected without cherry-picking. CHOrD effectively synthesizes diverse, spatially coherent layouts, while other methods produce various artifacts, including physically implausible object collisions, inconsistent object orientations, and missing objects. It should be noted that since all methods, including CHOrD, are data-driven, artifacts may persist due to the imperfect nature of the datasets themselves, such as those in 3D-FRONT discussed in Section 4. However, unlike CHOrD, other approaches produce notably more failure cases than those present in the training data. Figure 4 and Figure 7 demonstrate house-scale layouts synthesized by CHOrD, as well as photorealistic renderings. Other methods cannot generate house-scale layouts covering all rooms or irregular room shapes. Notably, CHOrD is able to generate diverse 2D layouts from the same floor plan, despite the CHOrD dataset containing only one layout per plan.

We present the quantitative evaluation of all methods in Table 2. Following previous work (Lin & Mu, 2024; Tang et al., 2024; Yang et al., 2024a), we use Frechet Inception Distance (FID) (Heusel et al., 2017) and Kernel Inception Distance (KID) (Bińkowski et al., 2018) to assess the quality and diversity of synthesized layout images. Additionally, we compute two metrics to evaluate bounding

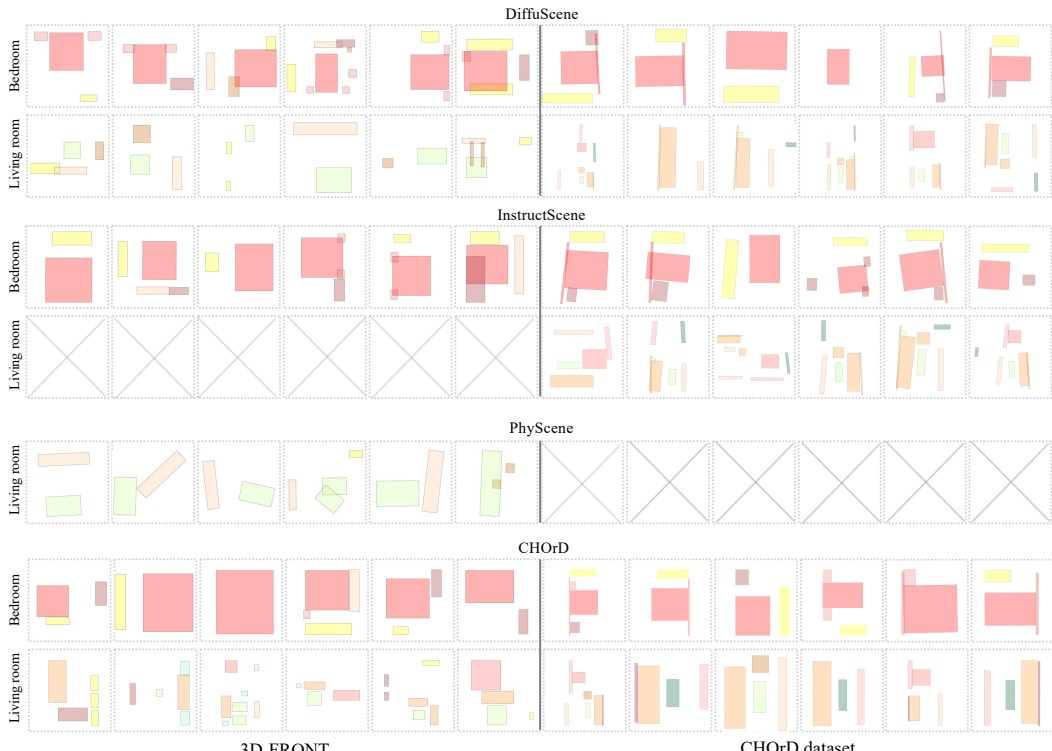

Figure 5: Visualization of synthesized layouts by CHOrD, DiffuScene (Tang et al., 2024), InstructScene (Lin & Mu, 2024), PhyScene (Yang et al., 2024a). All results were randomly selected from an arbitrary batch without any cherry-picking. It is evident that only CHOrD produces clean, coherent layouts, whereas others exhibit significant artifacts such as implausible overlapping items, inconsistent orientations, or missing objects.

box collisions in synthesized layouts: Pairwise Overlap Ratio (POR), quantifying the proportion of intersecting object pairs to the total number of pairs, and Pairwise Intersection over Union (PIoU), measuring the ratio of the intersecting area between two objects to the area of their union. The average values for these metrics are obtained by first computing per-scene values, followed by the arithmetic mean. CHOrD consistently achieves state-of-the-art performance across all metrics and datasets by a significant margin.

**Out-of-distribution (OOD) analysis** We aim to provide a theoretical explanation for why CHOrD is less likely than existing baselines to produce samples with spatial artifacts (*i.e.*, out-of-distribution samples). In diffusion models, the MSE loss for in-distribution and out-of-distribution samples is inversely correlated with likelihood (Ho et al., 2020), and can therefore serve as a reliable proxy for likelihood. A well-performing generative model should assign lower likelihood (higher MSE) to out-of-distribution samples than to in-distribution samples.

To evaluate whether out-of-distribution samples are correctly assigned lower likelihood (higher MSE), we use object collisions as an exemplary artifact to distinguish between in- and out-of-distribution samples and compute the MSE values of CHOrD and other baselines on these samples. Specifically, we use a set of 400 3D-FRONT samples with the largest PIoU values as out-of-distribution samples, and clean

| Method | w Col. (OOD) | w/o Col. |
|---|---|---|
| DiffuScene | 0.1877166 | 0.1877169 |
| InstructScene | 0.00316 | 0.00324 |
| **Ours** | **0.00071** | **0.00054** |

Table 3: MSE values of CHOrD and other methods on samples with and without collision.

3D-FRONT layout samples without collisions as in-distribution samples. The MSE is calculated by adding noise to the samples at timesteps 900–1000, measuring the mean squared error between the true and predicted noise, and averaging the results over 100 iterations. The same procedure is applied to all methods except PhyScene, whose training script is not publicly available.

Figure 6: Fine-grained coffee table and desk layouts that accommodate natural vertical object overlaps. The computer setup in the third column, consisting of a monitor, keyboard, and mouse, was modeled as a single object placed on the mat.

We present the MSE values of all methods for in- and out-of-distribution samples in Table 3. The results support our hypothesis: both InstructScene and DiffuScene, despite being trained on collision-free data, exhibit indistinguishable MSEs for samples with and without collisions, whereas CHOrD shows a significant 32.22% difference. This demonstrates the efficacy of CHOrD in capturing nuanced spatial patterns through an effective image-based layout representation.

## 5.2 FINE-GRAINED LAYOUT SYNTHESIS

As discussed in Section 3.2, the multi-level graph structure enables CHOrD to synthesize fine-grained layouts such as placing objects on a coffee table. This can be achieved by applying a separate conditional diffusion model, with the floor plan image conditioning replaced by an image indicating upper-level boundaries.

**Dataset and implementation**   Since neither the 3D-FRONT nor CHOrD datasets contain fine-grained layouts for this task, we additionally collected a small dataset of object placements on common household items such as dining tables, coffee tables, and desks. We recorded the object categories, positions, orientations, and sizes, as well as bounding boxes, and generated top-view images of their layouts. Object categorization and their color schemes are detailed in Appendix Table 5. The objects were drawn proportionally to their absolute sizes, with the maximum drawing area fixed at 2-meter squares. We adhered to the same training procedures as detailed in Section 5.1.

**Results**   We present exemplar results in Figure 6. CHOrD enables two mechanisms that prevent implausible object collisions while allowing natural vertical overlaps. First, as discussed in Section 3.2, the autoregressive multi-level layout generation allows fine-grained objects to be placed on upper levels, such as a computer on a desk. Second, some vertical overlaps do not exhibit clear hierarchical relationships, such as an object partially resting on a desk mat. In this scenario, we directly train the diffusion model with RGB images containing vertical overlaps, enabling it to generate plausible layouts with natural vertical overlaps while preventing unreasonable ones. The unique color assigned to each object guides the 2D diffusion model in distinguishing permissible overlaps from invalid ones. Due to the limited availability of natural partially overlapped objects, we demonstrate this feature only at the fine-grained level. Figure 6 illustrates both scenarios.

Implementation details and results of multi-modal floor planning are provided in Appendix D.

## 6 DISCUSSIONS AND SUMMARY

In this paper, we propose a novel framework that employs a 2D image-based intermediate layout representation to synthesize spatially coherent, house-scale, and hierarchically structured indoor 3D scenes. The success of CHOrD hinges on its comprehensively enhanced spatial understanding compared to existing solutions, such as tabular generative models or LLMs, which struggle to meet these objectives. We anticipate a series of intriguing applications of CHOrD in a variety of tasks such as interior design, virtual and augmented reality, and embodied AI simulation.

**Limitations**   CHOrD did not explore stylistic control of individual objects or text-guided object placement, as has been explored by prior works Lin & Mu (2024); Tang et al. (2024). However, as CHOrD can be integrated into these pipelines, we leave these features for future work. In extremely rare cases, YOLOv8 failed to detect precise bounding boxes, leading to misoriented objects or minor collisions despite the layout images being axis-aligned and collision-free. This can be readily addressed with more training data, thanks to the strong scalability of CHOrD, as evidenced in Appendix D. With the same amount of training data, CHOrD outperforms prior work by a significant margin.

## 7 ETHICS STATEMENT

In this work, we adhere to responsible research practices, ensuring that the datasets and methods used comply with legal and ethical standards. The dataset created and utilized in this study was sourced and generated without infringing on the rights of individuals or organizations. We have ensured that the data is free of personally identifiable information, and that no harm has been inflicted on any subjects during the research process. Furthermore, the research addresses technical challenges in generative indoor scene synthesis, with no direct societal or environmental risks.

## 8 REPRODUCIBILITY STATEMENT

To ensure the reproducibility of our work, we have provided the source code in the supplementary materials, along with detailed descriptions of our methodology and experimental setup in the main paper. The CHOrD model, including hyperparameters, training procedures, and evaluation metrics, has been thoroughly documented. Upon acceptance, we will publicly release the source code, pre-trained models, and our novel CHOrD dataset to facilitate verification and further research in this field.

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

# CHOrD: Synthesizing Spatially Coherent, House-Scale, Organized, and Diverse 3D Indoor Scenes via Image-Based Layout Guidance

## Appendix

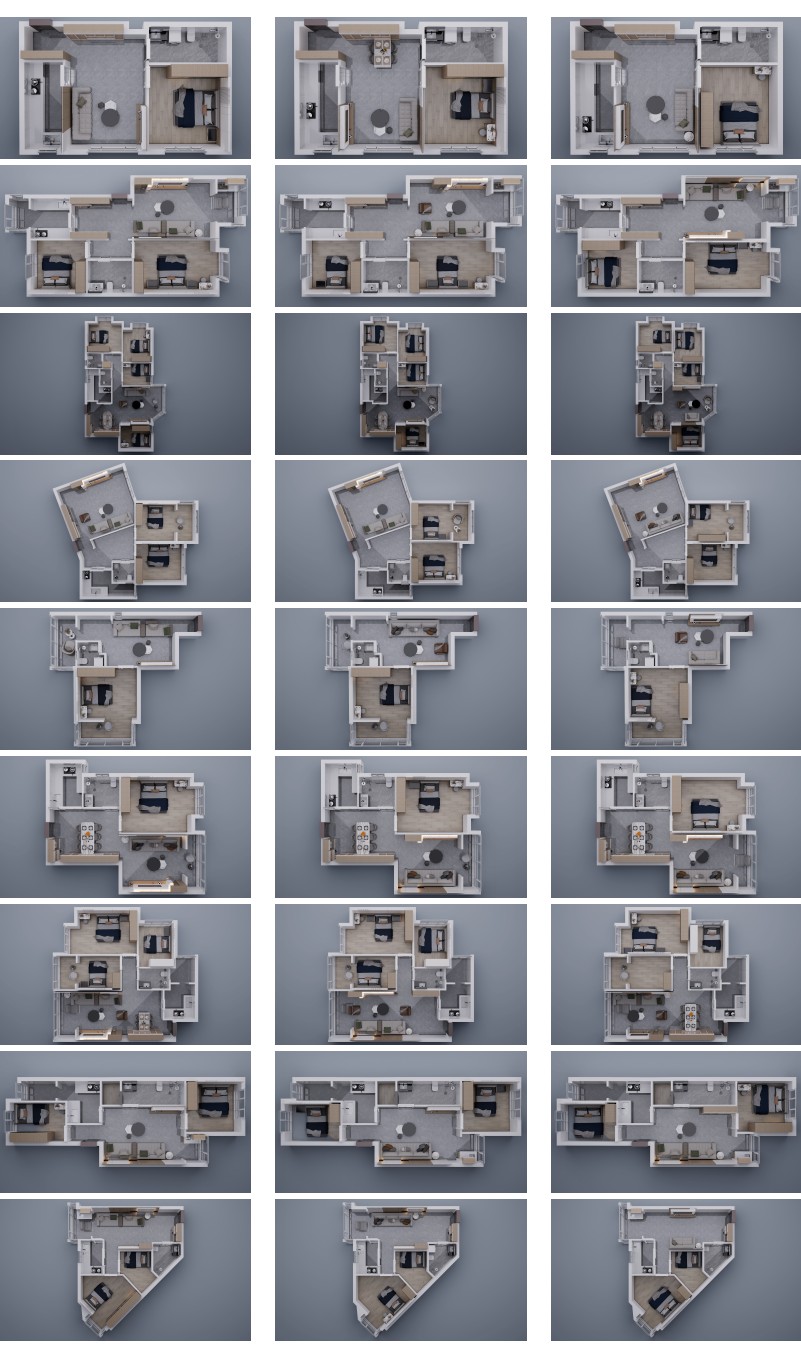

Figure 7: Photorealistic rendering of diverse synthesized layouts by CHOrD conditioned on complex floor plans.

## A  DENOISING DIFFUSION PROBABILISTIC MODEL BACKGROUND

We adopt the denoising diffusion probabilistic model (DDPM) to capture the distribution of the layout space. A diffusion model consists of two processes: a forward (diffusion) process and a reverse (generation) process.

**Forward process.** Given a data sample $x_0 \sim q(x_0)$, the forward process gradually corrupts $x_0$ by sequentially adding Gaussian noise through a Markov chain:

$$q(x_t|x_{t-1}) = \mathcal{N}\left(x_t; \sqrt{1-\beta_t}\, x_{t-1}, \beta_t I\right), \tag{3}$$

where $t \in \{1, \dots, T\}$, $\{\beta_t\}_{t=1}^{T}$ is a variance schedule, and $I$ is the identity matrix. With an appropriate schedule, the distribution $q(x_T)$ converges to a standard Gaussian. The training objective is to predict the injected noise:

$$L = \mathbb{E}_{x_0, \epsilon_t}\left[\left\|\epsilon_\theta(x_t, t) - \epsilon_t\right\|_2^2\right]. \tag{4}$$

**Reverse process.** The reverse process learns to invert the corruption by estimating a transition kernel from $x_t$ to $x_{t-1}$:

$$p_\theta(x_{t-1}|x_t) = \mathcal{N}(x_{t-1}; \mu_\theta(x_t, t), \Sigma_\theta(x_t, t)), \tag{5}$$

where $\theta$ are learnable parameters. Starting from a Gaussian prior $p(x_T) = \mathcal{N}(0, I)$, the data distribution can be approximated by marginalizing over all denoising steps:

$$p_\theta(x_0) = \int p(x_T) \prod_{t=1}^{T} p_\theta(x_{t-1}|x_t)\, dx_{1:T}. \tag{6}$$

## B  ADDITIONAL CHORD TECHNICAL DETAILS

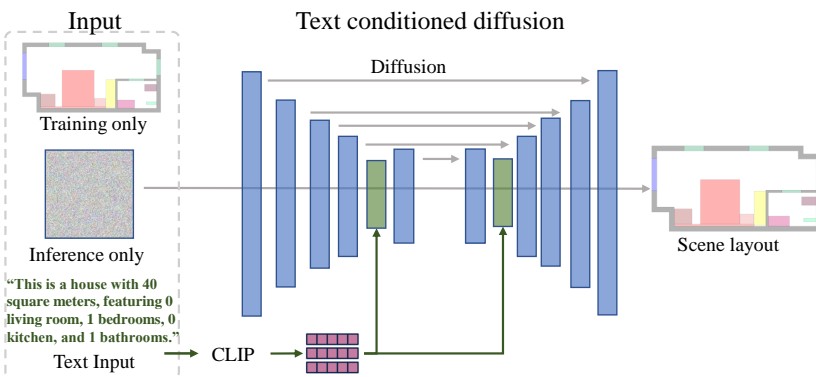

Figure 8: Text-conditioned diffusion model.

### B.1  MULTI-MODAL CONTROL DETAILS

Apart from the main pipeline, the 2D layout of CHOrD enables additional multi-modal controls for the floor plan and fine-grained layouts. Specifically, we provide three types of control:

**Text-conditioned floor planning** An alternative and convenient way to specify the floor plan is through natural language, especially when floor plan images are not accessible or incompatible with the format accepted by our model. Given the success of text-to-image diffusion models (Ramesh et al., 2022; Saharia et al., 2022b), text descriptions provide a viable alternative for floor plan specification, as shown in Figure 14. Specifically, as illustrated in Figure 8, we generate a fixed-size conditioning vector $y_c$ by passing the text input through a CLIP encoder (Radford et al., 2021).

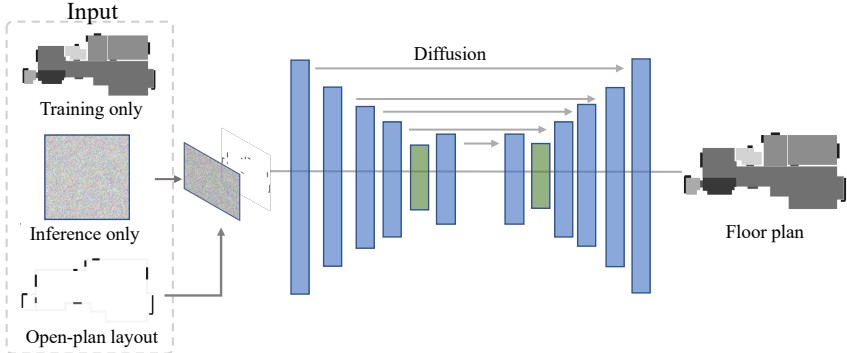

Figure 9: Open-plan-conditioned diffusion model.

Semi-structured text is particularly effective for this task (*e.g.*, "This is a 40-square-meter flat with 0 living rooms, 1 bedroom, 0 kitchens, and 1 bathroom."). The resulting CLIP embedding serves as the conditional variable for the diffusion model, guiding the generation of scene layouts based on high-level semantic information encoded in the text description. This allows for more intuitive control over the layout generation by leveraging natural language as an additional input modality. The text-based model is trained using the same loss as Equation 1, with the conditioning variable being the CLIP embedding $y_c$ instead of the floor plan image $y$. Conditioning is introduced through a cross-attention layer (Vaswani, 2017) near the UNet bottleneck.

**Open-plan-conditioned floor planning** CHOrD also supports synthesizing floor plans conditioned on an open-plan layout, as shown in Figure 15. Specifically, given a 2D image of an open-plan layout without room arrangements, CHOrD generates complete floor plans with optimal room separations. This is particularly useful for users looking to modify floor plan structures or synthesize digital twin environments with greater variety. As illustrated in Figure 9, the open-plan-conditioned diffusion model shares the same architecture as the floor plan-conditioned diffusion model detailed in Section 3.1, except that this model takes an open-plan figure as input and generates a structured floor plan with optimal room arrangements. The generated floor plan can then serve as input to the floor plan-conditioned diffusion model. In other words, open-plan-conditioned floor planning functions as an optional preprocessing step before the CHOrD main pipeline.

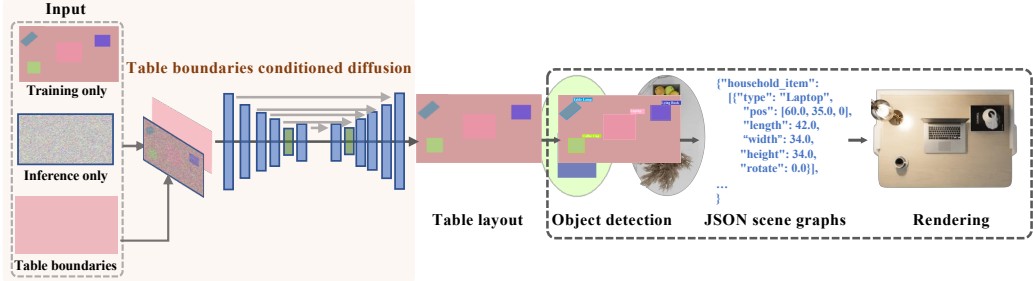

Figure 10: Overview of CHOrD on fine-grained layout generation for table.

**Boundaries-conditioned fine-grained layout generation** CHOrD also supports fine-grained layout generation conditioned on table boundaries. As illustrated in Figure 10, the diffusion model architecture and pipeline are similar to the main CHOrD pipeline, except that the conditional input is the table boundaries. Given a boundary image of a table, it can generate plausible object layouts, including items such as lamps, laptops, lying books, and coffee cups.

These controls are considered extended features of CHOrD—the main pipeline functions perfectly without them—but they are made possible largely due to our adoption of an image-based layout representation. We anticipate various additional features enabled by this approach.

| Category | Color | Category | Color |
|---|---|---|---|
| Bed | FF0000 | Cabinet | FFFF00 |
| Bed Background | FF3333 | Bedside Table | F08080 |
| Table | A52A2A | Leisure Sofa | 666600 |
| Sofa | FF9933 | TV Cabinet | FFCC99 |
| Sofa Background | 99004C | Coffea Table | CCFF99 |
| Dining Cabinet | FF9999 | Shoe Cabinet | 006633 |
| Single Sofa | CC6600 | Dining Table | FF6666 |
| Side Coffea Table | 99FFCC | Single Door Floor Cabinet | 9999FF |
| Double Door Floor Cabinet | 6666FF | Cooker Cabinet | 000099 |
| Sink Cabinet | 0000CC | Electrical FLoor Cabinet | 3333FF |
| Refrigerator | 006666 | Shower | 33FF99 |
| Toilet | 660033 | Washbasin | CC0066 |
| Washing Machine | FFCCE5 | Washing Set | FF66B2 |
| Wall | 000000 | Door | 139C5A |
| Window | 0000FF | | |

Table 4: Scene layout items and corresponding color schemes, with the opacity level set to 0.3.

## B.2 Rendering Details

We have preconfigured multiple sets of material style templates from an aesthetic perspective, including styles such as modern, light luxury, and vintage. These templates include a variety of items, such as beds and sofas of different sizes, flooring, wall paint, lights, decorations, and cabinets. After generating the positions and sizes of major objects using CHOrD, we match the objects to the most suitable items in the template based on the room type and dimensions. Simultaneously, we match appropriate flooring, wall paint, and lights from predefined material templates to the room type. For instance, bedrooms are matched with wooden flooring and wall paint, while bathrooms and kitchens are matched with tiles. This ensures not only a consistent furniture style but also alignment between the furniture and the flooring, wall paint, and lighting styles.

As rendering is not a core part of the CHOrD pipeline in terms of novelty, we only briefly mentioned it in the main paper, but it is essential in the final presentation of results. In the paper, we have rendered all the scenes using the modern style template, as we leave style control to future work.

| Category | Color | Category | Color |
|---|---|---|---|
| Bedside Table | F08080 | Table | A52A2A |
| Coffea Table | CCFF99 | Side Coffea Table | 99FFCC |
| Dining Table | FF6666 | Lying Book | 0000FF |
| Standing Book | FFFFAA | Magazine | 7FFFAA |
| All-in-one Computer | 00FFAA | Laptop | FF7FAA |
| Big Mouse Pad | 7F7FAA | Table Lamp | 007FAA |
| Small Ornament | FF00AA | Pen Holder | 7F00AA |
| Big Plant | 0000AA | Small Plant | FFFF55 |
| Coffee Cup | 7FFF55 | Electronic | FF0000 |
| Photo Frame | FF7F55 | Food | 7F7F55 |
| Dinner Set | FFFF00 | Drinks | 7F7F00 |

Table 5: Fine-grained items and corresponding color schemes.

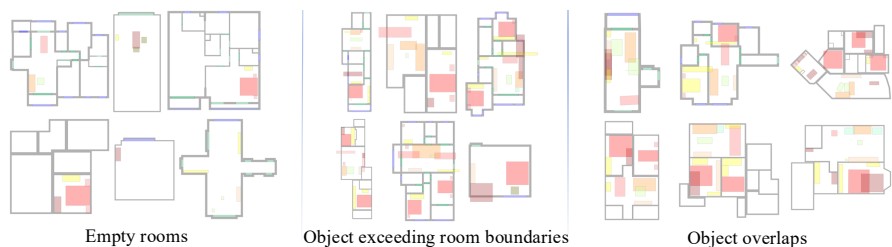

Empty rooms     Object exceeding room boundaries     Object overlaps

Figure 11: Erroneous scenes in 3D-FRONT.

## C  ADDITIONAL CHORD DATASET DETAILS

CHOrD dataset offers several clear advantages over 3D-FRONT:

**Expanded coverage of household items and room categories**  While 3D-FRONT provides instance semantic labels for 34 categories and 10 super-categories of household items, its dataset primarily includes objects placed in living rooms, bedrooms, and dining rooms, with no objects for kitchens, bathrooms, or balconies. Consequently, the layouts in 3D-FRONT are consistently devoid of furnishings in these areas, as seen in Figure 11. Our CHOrD dataset fills this gap by offering 26 super-categories of household items, including furniture, fixtures, and appliances, that comprehensively cover living rooms, bedrooms, dining rooms, kitchens, bathrooms, and balconies. Our CHOrD dataset not only contains more valid living rooms and bedrooms (each with at least one household item in place), but also includes outfitted kitchens and bathrooms.

**Improved data quality**  As frequently reported Zhang et al. (2018); Ritchie et al. (2019); Paschalidou et al. (2021); Tang et al. (2024); Lin & Mu (2024), the 3D-FRONT dataset contains erroneous layouts such as empty rooms, unnatural object sizes, misclassified items, and unrealistic object placements (*e.g.*, furniture outside room boundaries, lamps on the floor, blockage of doorways, and overlapping objects), as seen in Figure 11. Consequently, previous work (Zhang et al., 2018; Ritchie et al., 2019; Paschalidou et al., 2021; Tang et al., 2024; Lin & Mu, 2024) using 3D-FRONT invested considerable effort in data cleaning, removing numerous layouts with artifacts, which greatly reduced the amount of valid data. In contrast, our dataset is ready to use without these artifacts.

An example CHOrD data stored in JSON format is shown in List 1. A comprehensive statistic of the CHOrD dataset in comparison with 3D-FRONT is detailed in Table 7, 8, 9, and Figure 12.

Listing 1: Example JSON data format

```
{
  "rooms": [
    {
      "roomId": "D5F19A0446724E",
      "roomName": "living", # inner room
      "roomType": 1,
      "wallPoints": [
        [171.65, 241.5],
        [651.66, 241.5],
        ...] # 2d coords
    },
    {
      "roomId": "D5F19A044672",
      "roomName": "out_room",
      "roomType": 0,
      "wallPoints": [
        [171.65, 241.5],
        [651.66, 241.5],
        ...] # 2d coords
    }],
  "windowsDoors": [
```

```
    {
      "type": "door",
      "pos": [717.32, 737.0, 0],
      # box center position x,y;
      # height to floor z
      "length": 95,
      "width": 12,
      "height": 210,
      "rotate": 100
      # roate angle in degree
    },
    {
      "type": "window",
      "pos": [657.66, 945.12, 90],
      "length": 153.75,
      "width": 12,
      "height": 110,
      "rotate": 90.0
    }
    ],
  "furniture": [
    # 3d bounding box data,
    # same with windows and doors
    {
      "type": "coffee_table",
      "pos": [569.91, 1844.75, 0],
      "length": 76.0,
      "width": 94.0,
      "height": 99,
      "rotate": 180.0
    },
    {
      "type": "sofa",
      "pos": [411.66, 169.45, 0],
      "length": 185.3,
      "width": 120.1,
      "height": 99,
      "rotate": 0.0
    }
    ]
}
```

| Item | Category |
|------|----------|
| Nightstand | bedside table |
| Wardrobe | cabinet |
| Three-Seat / Multi-seat Sofa | sofa |
| Dining Table | dining table |
| Coffee Table | coffee table |
| Loveseat Sofa | sofa |
| Children Cabinet | cabinet |
| Drawer Chest / Corner cabinet | cabinet |
| King-size Bed | bed |
| TV Stand | tv cabinet |
| Sideboard / Side Cabinet / Console | dining cabinet |
| Lazy Sofa | leisure_sofa |
| Dressing Table | table |
| Wine Cabinet | dining cabinet |
| L-shaped Sofa | sofa |
| Corner/Side Table | side coffee table |
| Bookcase / jewelry Armoire | cabinet |
| Kids Bed | bed |
| Sideboard / Side Cabinet / Console Table | table |
| Bed Frame | bed |
| Shoe Cabinet | shoe cabinet |
| Three-Seat / Multi-person sofa | sofa |
| Double Bed | bed |
| Bunk Bed | bed |
| Desk | table |
| Two-seat Sofa | sofa |
| Tea Table | coffee table |
| Couch Bed | bed |
| Single bed | bed |
| Chaise Longue Sofa | sofa |
| U-shaped Sofa | sofa |

Table 6: 3D-FRONT furniture items and remapped categories.

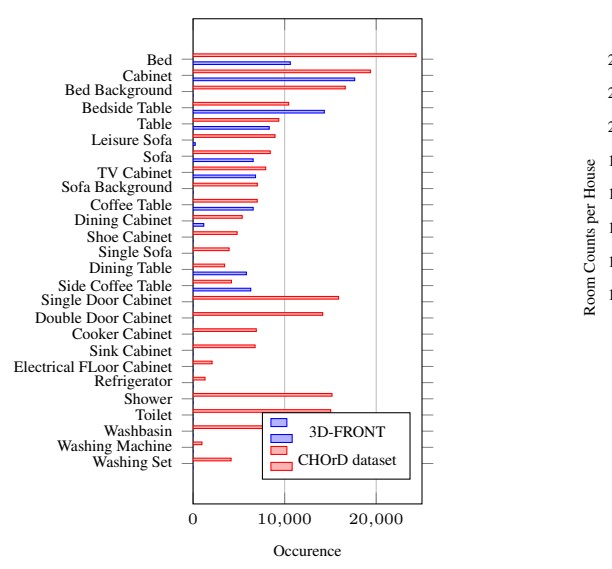

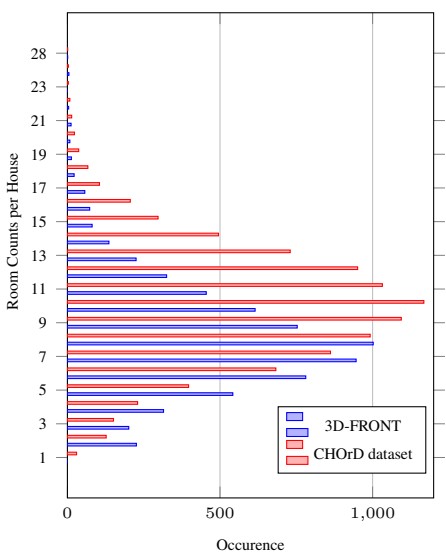

(a) Distribution of household item occurrences per super-category.

(b) Distribution of room counts per house, with an average of 9.78 and a total of 94,964 counts.

Figure 12: Statistics of the CHOrD dataset in comparison with 3D-FRONT.

| Room | Furniture | 3D-FRONT | CHOrD dataset (ours) |
|------|-----------|----------|----------------------|
| Bedroom | Bed | 10620 | 24354 |
| | Cabinet | 17649 | 19365 |
| | Bed Background | 0 | 16619 |
| | Bedside Table | 14333 | 10439 |
| | Table | 8318 | 9359 |
| Living Room | Leisure Sofa | 237 | 8953 |
| | Sofa | 6564 | 8430 |
| | TV Cabinet | 6821 | 7935 |
| | Sofa Background | 0 | 7019 |
| | Coffee Table | 6565 | 7005 |
| | Dining Cabinet | 1169 | 5368 |
| | Shoe Cabinet | 0 | 4817 |
| | Single Sofa | 0 | 3939 |
| | Dining Table | 5822 | 3444 |
| | Side Coffee Table | 6300 | 4195 |
| Kitchen | Single Door Cabinet | 0 | 15889 |
| | Double Door Cabinet | 0 | 14156 |
| | Cooker Cabinet | 0 | 6904 |
| | Sink Cabinet | 0 | 6773 |
| | Electrical Cabinet | 0 | 2081 |
| | Refrigerator | 0 | 1307 |
| Bathroom | Shower | 0 | 15174 |
| | Toilet | 0 | 15026 |
| | Washbasin | 0 | 12517 |
| | Washing Machine | 0 | 970 |
| Balcony | Washing Machine Cabinet | 0 | 4153 |

Table 7: Comparison of object occurrences between 3D-FRONT and CHOrD dataset.

| | Empty Room Rate | POR | PIoU |
|---|---|---|---|
| 3D-FRONT | 0.5906 | 0.0361 | 0.2547 |
| CHOrD dataset (ours) | 0.2902 | 0.0044 | 0.0018 |

Table 8: Comparison of data quality statistics between 3D-FRONT and CHOrD dataset.

| | Living | Bedroom | Kitchen | Bathroom | Balcony |
|---|---|---|---|---|---|
| 3D-FRONT | 1813 | 4041 | 0 | 0 | 0 |
| CHOrD dataset (ours) | 15115 | 40983 | 8262 | 16351 | 8262 |

Table 9: Comparison of non-empty room statistics between 3D-FRONT and CHOrD dataset.

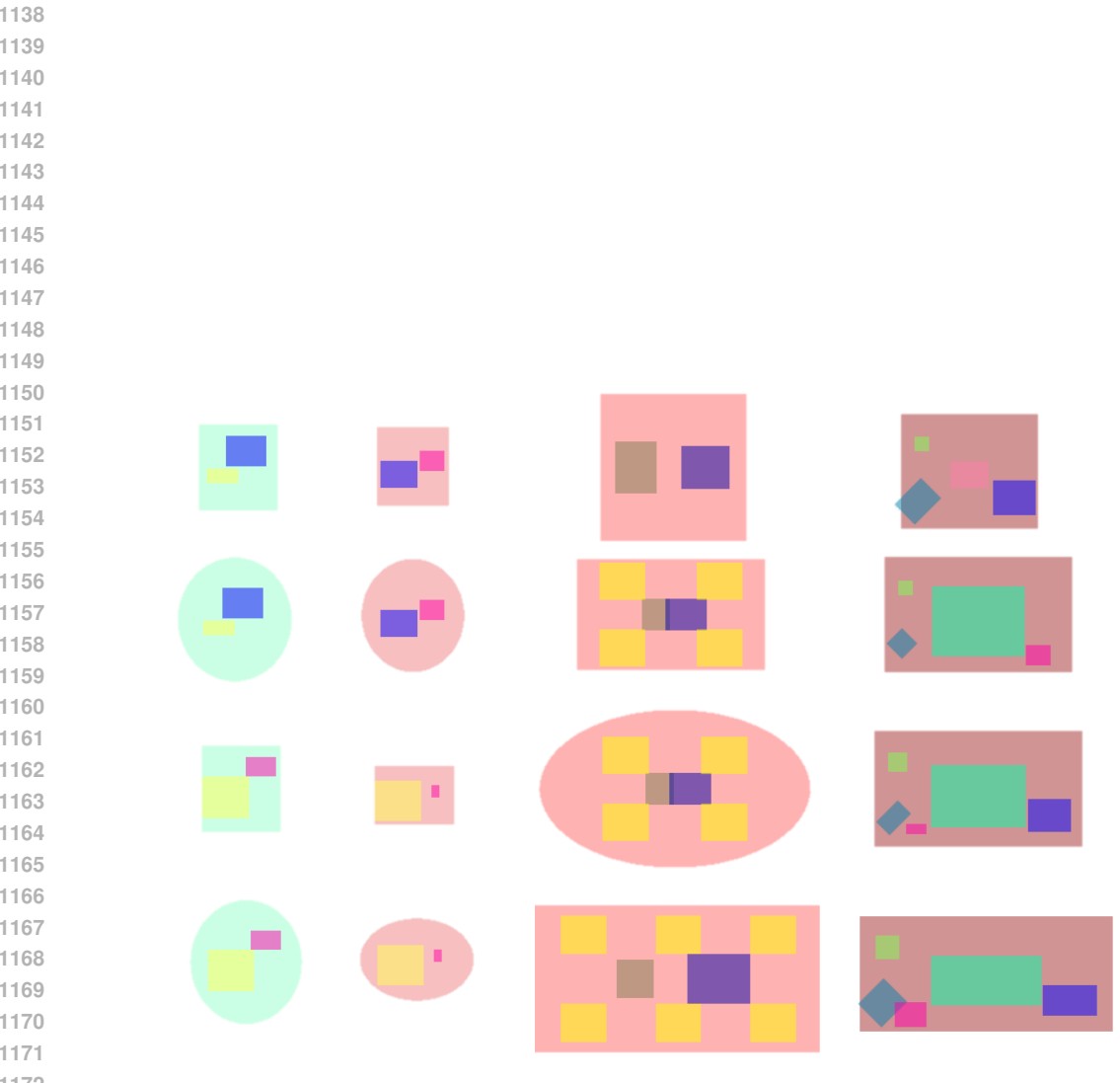

Figure 13: Fine-grained layout synthesis.

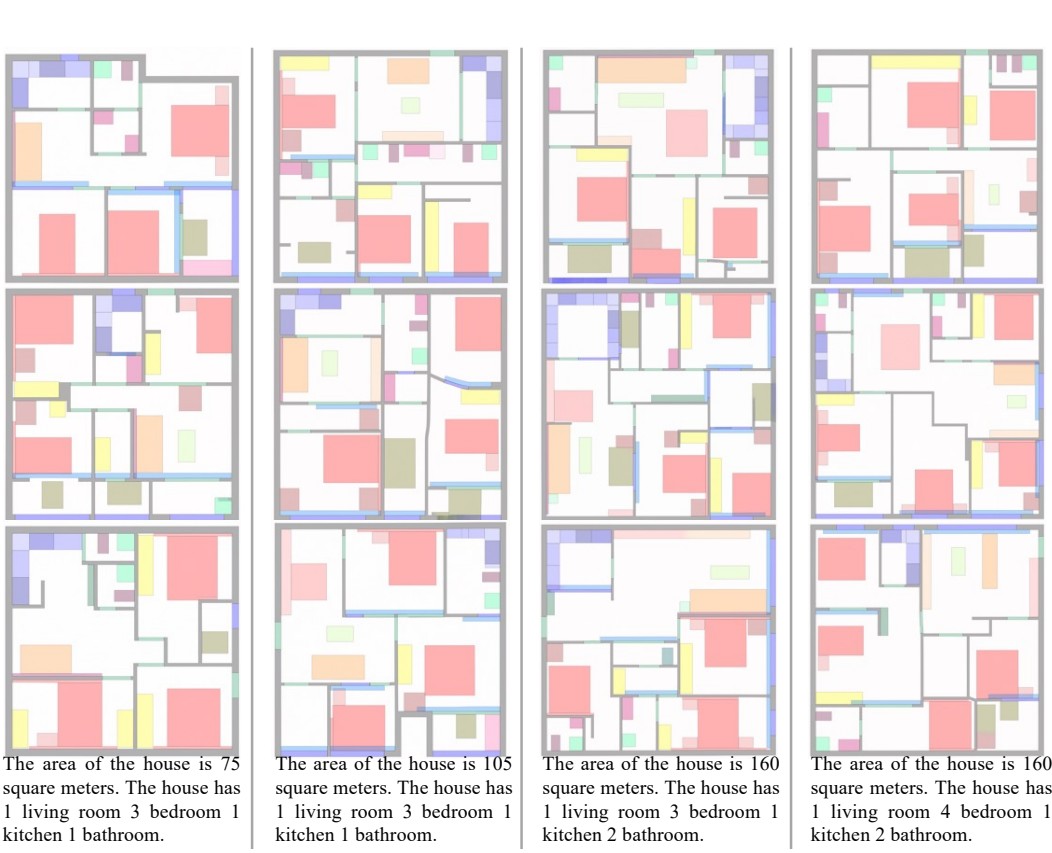

The area of the house is 75 square meters. The house has 1 living room 3 bedroom 1 kitchen 1 bathroom.

The area of the house is 105 square meters. The house has 1 living room 3 bedroom 1 kitchen 1 bathroom.

The area of the house is 160 square meters. The house has 1 living room 3 bedroom 1 kitchen 2 bathroom.

The area of the house is 160 square meters. The house has 1 living room 4 bedroom 1 kitchen 2 bathroom.

Figure 14: Visualization of text-to-layout generation by CHOrD trained on our CHOrD dataset. Floor plans of different room sizes all fill the entire canvas, with the wall thickness set to 24 cm for all scenes. Hence, the room size can be inferred from the thickness of the gray walls, which is consistent with the raw training data.

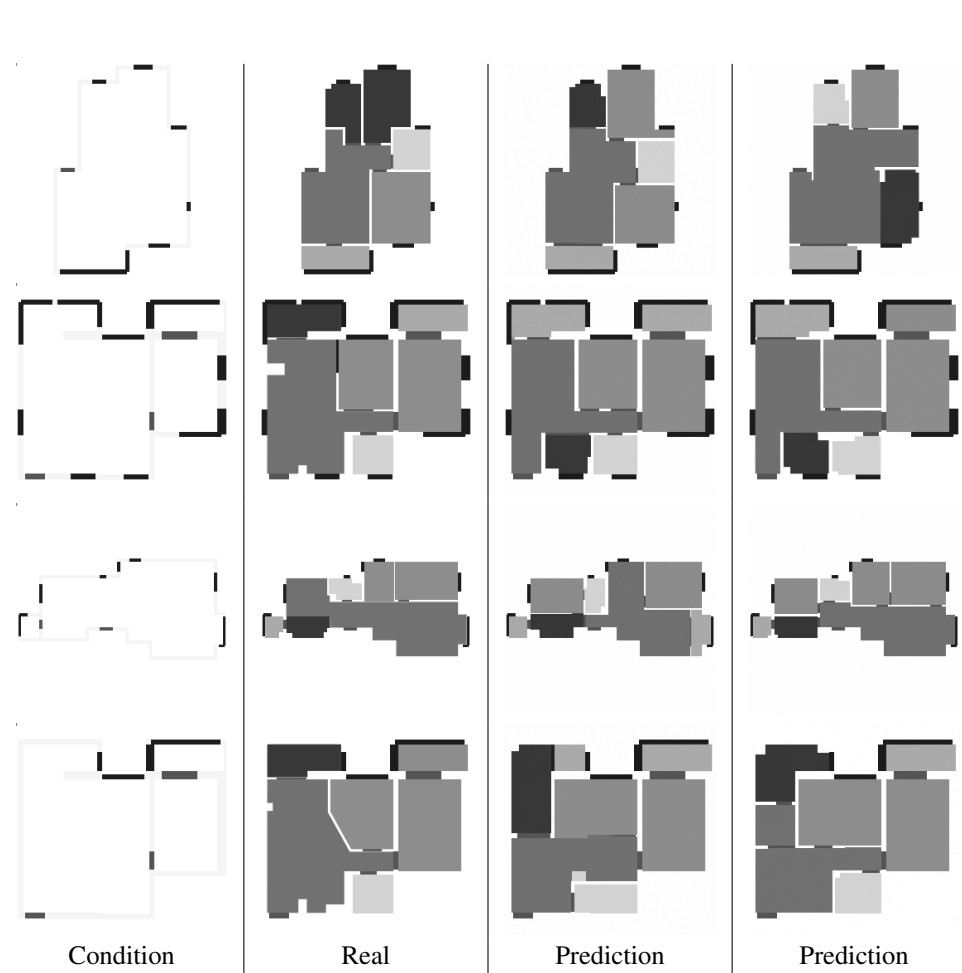

Condition    Real    Prediction    Prediction

Figure 15: Open-plan-conditioned floor planning.

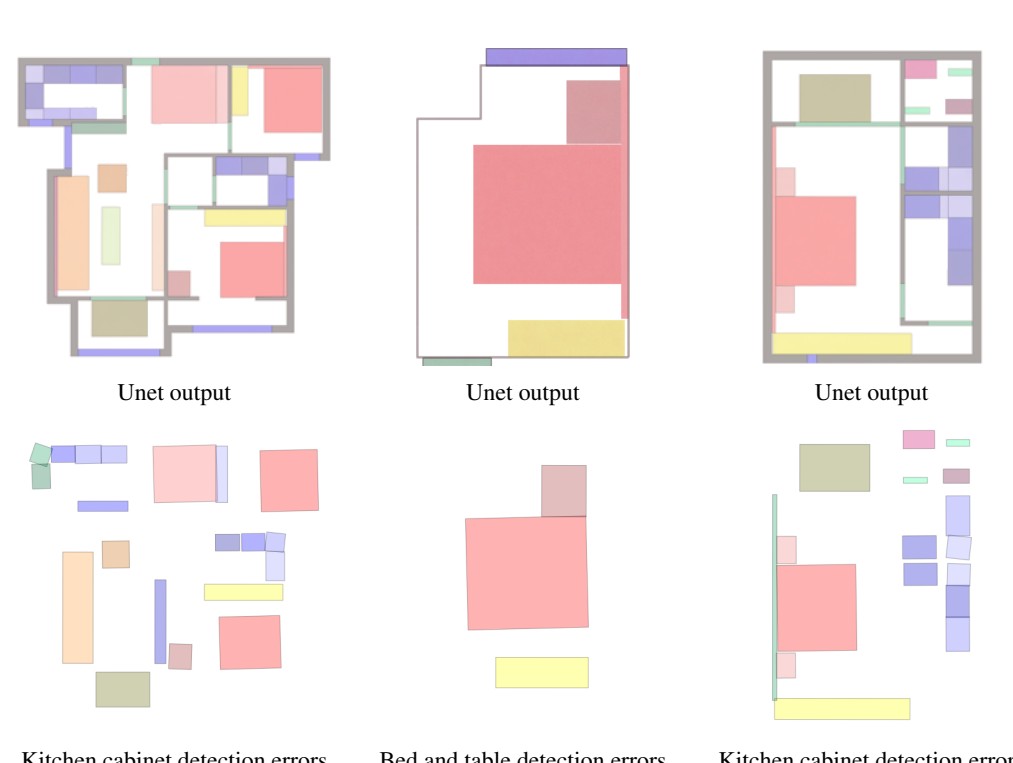

Figure 16: Sporadic failure cases due to YOLO detection errors when trained with insufficient data.

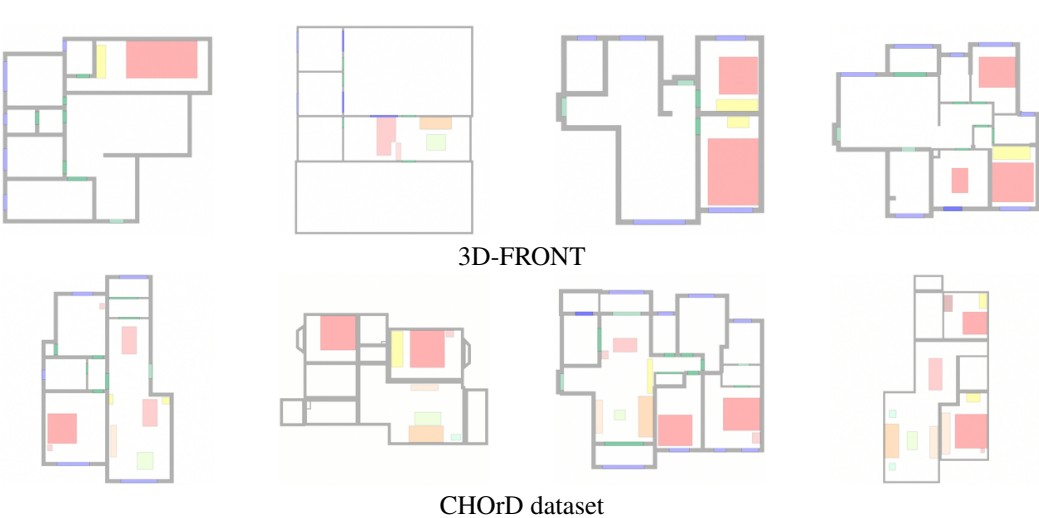

Figure 17: Performance of CHOrD on 3D-FRONT and CHOrD dataset, where results obtained from training on 3D-FRONT exhibit implausible unfurnished rooms due to artifacts in the original database.

# D    ADDITIONAL CHORD IMPLEMENTATION DETAILS AND EXPERIMENTAL RESULTS

**CHOrD inference efficiency**    On a single RTX 8000 GPU, the diffusion model inference takes approximately 9 seconds and 20 steps, enabled by DPMSolver (Lu et al., 2022). YOLO object detection takes around 40 milliseconds. 3D model matching and scene construction take about 100 milliseconds. Rendering, performed using the UE engine (Epic Games), takes approximately 30 seconds for a 2K image and around 120 seconds for a 4K image. While rendering is the most time-consuming module, it is an independent component that can be flexibly replaced with any real-time rasterization-based renderer when efficiency is a priority. We chose a ray-tracing-based renderer for photorealistic quality.

**Multimodal control implementation and results**    For text-conditioned floor planning, we parse the JSON file of each scene in the CHOrD dataset to extract the total area, room count, and categories to generate the corresponding textual description. For open-plan-conditioned floor planning, we use the CHOrD dataset to generate open-plan layouts and floor plans with proper room arrangements as grayscale images, with different colors representing room types. Both experiments followed the same training procedures as detailed in Section 5.1.

We present additional qualitative results for fine-grained layout synthesis in Figure 13, text-conditioned floor planning in Figure 14, open-plan-conditioned floor planning in Figure 15.

**Scalability and generalizability analysis**    In rare instances, YOLOv8 struggled to detect accurate bounding boxes, resulting in misaligned objects or minor collisions, even though the layout images were axis-aligned and collision-free, as shown in Figure 16. We demonstrated that this can be straightforwardly addressed with more training data. Specifically, we trained CHOrD on a privately collected dataset of over 100,000 indoor scenes, achieving significantly better results (**FID** 17.76, **KID** 0.02, **POR** 0.005, **PIoU** $4.399 \times 10^{-5}$) with substantially fewer failure cases compared to the results obtained from training on the CHOrD dataset (9,706 scenes) and reported in Table 2. CHOrD also performs considerably better when trained on CHOrD dataset compared to 3D-FRONT, as illustrated in Figure 17. These results evidence the strong scalability and generalizability of CHOrD.

# E    USE OF LARGE LANGUAGE MODELS

LLMs were used sparingly to polish the writing of this paper, primarily for checking grammar and typos.

