# OpenReview forum: "CHOrD: Synthesizing Spatially Coherent, House-Scale, Organized, and Diverse 3D Indoor Scenes via Image-Based Layout Guidance"
_ICLR.cc/2026/Conference — Submitted to ICLR 2026_

### Official Review · Reviewer_gk1H · 2025-10-27

**Soundness:** 4
**Presentation:** 3
**Contribution:** 3
**Rating:** 8
**Confidence:** 4

**Summary:**

The paper introduce CHOrD, a two-stage generative framework for indoor house-level scene synthesis and digital twin creation. The main novel contribution of this work stems from the fact that instead of directly predicting a 3D scene graph or object list, CHOrD first generates a 2D image-based furnished scene layout conditioned on a given floor plan image using a diffusion-based image-to-image model. This output is then parsed by a fine-tuned YOLOv8 object detector and segmenter to extract a hierarchical scene graph. 3D meshes are retrieved and rendered according the scene graph to produce photorealistic, simulation-ready environments.

The core insight of the authors is that an intermediate 2D layout produced by image encoders and decoders with strong spatial priors enhances spatial reasoning, allowing CHOrD to effectively avoid common artifacts such as object collisions, misalignment, and out-of-bound placements—without the need for costly post-processing of collision checks or iterative self-correction. In addition, CHOrD can support multi-level autoregressive layout generation, enabling fine-grained spatial composition, e.g., objects on tables, and multi-modal conditioning, e.g., text-guided and open-plan floor planning.

The authors also present a new CHOrD dataset containing 9,706 clean, fully furnished scenes covering 26 furniture super-categories, including kitchens, bathrooms, and balconies - areas underrepresented in prior datasets. The quantitative results provided in the paper on both 3D-FRONT and CHOrD datasets show state-of-the-art performance compared to baselines, while qualitative comparisons confirm CHOrD’s spatial coherence and robustness to irregular room shapes.

**Strengths:**

The paper's core innovation is the introduction of an image-based intermediate layout representation as a key insight, which significantly enhances spatial reasoning and coherence. This approach to scene graph generation effectively reduces spatial artifacts, while being robust to implementation in various other pipelines, enabling streamlined adoption to the industry.

The paper’s central claim—the use of a generative model to produce an image-based intermediary representation leveraging strong spatial priors—is clearly articulated and well supported. The intuition behind this design choice is effectively explained and further substantiated through both experimental analysis and qualitative comparisons to baseline methods. The authors provide quantitative results across multiple datasets and evaluation metrics (FID, KID, POR, PIoU), demonstrating consistent superiority over baselines.

The paper presents a clear motivation and problem definition, addressing the challenge of spatial incoherence in 3D indoor scene synthesis. Its core insight—introducing a 2D image-based layout as an intermediate representation— enhances spatial reasoning and coherence. CHOrD is fully data-driven, avoiding handcrafted rules, collision detection, or iterative regeneration. Moreover, CHOrD supports hierarchical and fine-grained layout generation, enabling realistic multi-level spatial relationships.

The authors also contribute a high-quality dataset that expands room and object coverage beyond 3D-FRONT, and provide a comprehensive evaluation demonstrating consistent state-of-the-art performance across multiple metrics and datasets.

Finally, the model shows robustness to out-of-distribution spatial artifacts, supported by both theoretical justification and empirical validation.

**Weaknesses:**

The paper would benefit from comparisons to a broader range of benchmarks. CHOrD is effectively compared only to InstructScene and DiffuScene, with the comparison to PhyScene being quite limited. Although other baselines did not release training code, it would still strengthen the paper to include reported metrics from those works, even with appropriate caveats, to better contextualize CHOrD’s performance within the broader literature.

The paper lacks ablation studies. For instance, the necessity and impact of training the diffusion model are not analyzed or discussed, nor is the contribution of the segmentation component examined in detail.

The discussion of CHOrD’s limitations is somewhat superficial. The section primarily lists unaddressed directions—such as stylistic control or segmentation accuracy—but does not critically analyze inherent limitations of the proposed method itself, nor does it present or reflect on any observed failure cases.

Minor issues:

Some phrases are repeated unnecessarily — for instance, the example “such as placing objects on a coffee table” appears three times throughout the paper.

Figure 4: The highlighted squares do not accurately correspond to the visible regions, and it would be helpful to indicate the camera orientations. Additionally, the ordering of the images on the right side seems arbitrary and would benefit from a clearer logical structure.

Section 3.2 (Fine-grained Layout): The paragraphs might flow better if presented in reverse order.

Tables 1 and 2: The method names should be consistent across both tables—currently, one lists reference names while the other uses method names. Moreover, Table 2 is missing a “Method” column title, which should be added for clarity.

Section 3.1: the sentence "In 2D images, implausible spatial artifacts are instantly
visible and flagged as OOD samples, enabling the model to generate coherent, realistic layouts." requires further explanation or citation.

**Questions:**

Have you evaluated how much each component (e.g., diffusion model, YOLO segmentation) contributes to performance? How was the amount of training determined?

Were any stability or convergence issues observed during diffusion training?

Have you evaluated the performance of CHOrD across datasets (e.g., trained on 3D-FRONT and tested on CHOrD or vice-verse) to validate robustness of training?

What types of room layouts or object arrangements provide a challenge for CHOrD?

How does CHOrD handle non-orthogonal geometries such as circular rooms? Some examples on this would be good.
Additionally, is CHOrD able to position furniture such that it is not aligned with any of the walls?

---

> ### Author Response · Authors · 2025-11-27
>
> We thank the reviewer for the thorough and constructive feedback. We are glad that you found the core insight, motivation, and spatial coherence of CHOrD to be clearly articulated, and we appreciate your positive assessment of the contribution, soundness, and dataset quality. Below we address all concerns in detail.
>
> ## Comparisons to a broader range of baselines
> We agree that broader comparisons are valuable. Unfortunately, many recent methods, including LayoutGPT, Holodeck, COFS, Forest2Seq, DEBARA, and SceneFactor, did not release training code and therefore cannot be fine-tuned on 3D-FRONT/CHOrD for consistent quantitative comparison.
>
> While their original papers report partial metrics, these numbers are not directly comparable, as each method employs a different rendering protocol; consequently, the reported FID/KID values are unfortunately not compatible across methods.
>
> Various spatial artifacts, such as collisions and out-of-bound objects, have also been reported in the original papers of these methods (e.g., Holodeck, LayoutGPT, etc.)
>
> ## Ablation studies
> We appreciate the suggestion. As CHOrD is a procedural pipeline, it requires the output of each previous step to serve as the input for the next. Therefore, it does not naturally lend itself to ablations where an entire component can be removed to study its necessity without breaking the pipeline.
>
> In principle, the YOLO detector could be replaced by non-learning-based image processing methods. However, as the diffusion model produces images with noise rather than perfectly clean color blocks, object attributes cannot be reliably determined solely from pixel colors. A detection module is therefore needed to robustly parse object categories, positions, and orientations from the generated layouts.
>
> ## Limitations discussion
> Thank you for the suggestion. We are fully transparent about our limitations. As discussed in Section 6 and shown in Figure 16, CHOrD is reliant on the detection accuracy of YOLO. In occasional cases, YOLOv8 fails to produce precise bounding boxes, which can lead to misoriented objects or minor collisions, even when the underlying layout image is axis-aligned and collision-free.
>
> This issue can be mitigated by incorporating more training data, as reported in Appendix D (L1369–1377). Under the same amount of training data, CHOrD indeed outperforms prior work by a significant margin.
>
> ## Minor issues
> Thank you for pointing these out. We will correct these issues in the revision.
>
> The sentence regarding OOD samples is further explained in Section 5 (L415-445).
>
>
> ## Q: Contribution of components; training amount
> As noted above, since CHOrD is a procedural pipeline in which each stage relies on the output of the previous one, it does not naturally lend itself to ablations where an entire component can be removed to study its contribution, without breaking the pipeline.
>
> That said, we are happy to include additional ablation studies in the revision if the reviewer could specify the exact ablation settings or comparison groups they would like to see.
>
> The amount of training was determined by following the default configurations of DDPM (Ho et al., 2020) and YOLOv8 (Jocher et al., 2023), both of which converge nicely for our task. Training and inference details are provided in L330–335 and Appendix D.
>
> ## Q: Training stability or convergence issues
> We did not observe any significant stability issues during training. DDPM training was stable across all runs using standard noise schedules, and DPM-Solver further stabilizes inference. The model typically converged within the first ~100 epochs without noticeable oscillations or divergence on either dataset.
>
>
>
> ## Q: Cross-dataset robustness
> We have now performed this experiment and report the cross-dataset evaluation below:
>
> | Training Setting                     | FID          | KID           | POR          | PIOU         |
> |--------------------------------------|--------------|---------------|--------------|--------------|
> | Train on 3D-FRONT, eval on ours      | 83.38780923  | 0.122821572   | 0.020452433  | 0.000468338  |
> | Train on ours, eval on 3D-FRONT      | 56.45894305  | 0.064099704   | 0.01745261   | 0.001048715  |
>
> However, we would like to note that the interpretability of FID/KID in this cross-dataset setting is limited. For instance, models trained on CHOrD dataset tend to include more furniture, which is both qualitatively and quantitatively preferable when evaluated against the CHOrD dataset itself. However, this can lead to higher FID when evaluated against 3D-FRONT due to **distributional mismatch**, despite producing better layouts.
>
> We will include these quantitative results in the revised version.

---

> ### Author Response · Authors · 2025-11-27
>
> ## Q: Challenging room/scene types
>
> Like any ML system, corner cases with very limited training data are the most challenging scenarios.
>
> Large rooms, particularly when they have unseen geometries, are inherently more difficult due to their relative scarcity in the data. A potential solution is to augment the training set with standard transformations such as mirroring and rotation.
>
> ## Q: Non-orthogonal geometries (circular/angled rooms)
> Our dataset unfortunately does not contain circular rooms. However, as CHOrD employs an image-based representation, the model is not inherently restricted by room shape. Given sufficient training data, it should theoretically generalize to circular geometries as well.
>
> CHOrD has already shown robustness to rotations and to irregular or slanted room shapes. For example, in Figure 4 (first row) and Figure 7 (fourth and last rows), the placements of objects remain correctly oriented and consistently aligned with rotated or angled wall geometries.
>
> The difficulty in these cases arises not from the geometry itself, but from the limited number of such examples relative to standard, axis-aligned rooms.
>
> ## Q: Positioning objects not aligned with walls
> CHOrD is purely data-driven. Orientation alignment is **not** a hand-designed rule but learned directly from the data. If the training data largely contains alignments parallel to walls, the model naturally learns this pattern. Conversely, if arbitrary orientations are permitted in the data, CHOrD will also produce arbitrarily oriented placements.
>
> For instance, in fine-grained table layouts (Figure 6; Figure 13, last column), the objects exhibit diverse rotations because the underlying dataset does not impose any particular alignment constraint.

---

### Official Review · Reviewer_v9NZ · 2025-10-29

**Soundness:** 3
**Presentation:** 3
**Contribution:** 2
**Rating:** 2
**Confidence:** 5

**Summary:**

This paper introduces CHOrD, a generative framework for synthesizing spatially coherent, house-scale, and hierarchically organized 3D indoor scenes. The core innovation is a two-stage process that first generates an intermediate 2D image-based layout representation from a floor plan, which is then converted into a scene graph. This approach leverages the spatial reasoning capabilities of image-based models to mitigate common artifacts like collisions and incomplete layouts observed in prior tabular or LLM-based methods. The authors also introduce a new, high-quality dataset, the CHOrD dataset.

**Strengths:**

- Unlike many prior methods restricted to simplistic rectangular rooms or single-room layouts, CHOrD can handle house-scale layouts with complex, irregular room shapes and floor plan structures.

- The paper introduces a large, clean dataset (9,706 scenes, $\approx 1.4\times$ larger than 3D-FRONT). This dataset offers expanded coverage (including fully furnished kitchens, bathrooms, and balconies) and is artifact-free.

- CHOrD achieves superior quantitative results across all key metrics (FID, KID, POR, PIoU) on both the 3D-FRONT and the proposed CHOrD datasets, demonstrating its ability to generate high-quality, diverse, and coherent layouts.

**Weaknesses:**

- The pipeline's second stage relies on fine-tuned YOLOv8 to detect objects and extract the structured scene graph from the generated 2D image. How to ensure the object orientation within the scene graph, since the orientation of object is very important for a reasonable scene structure. Can you quantify the failure rate of YOLOv8 detection and describe the specific training strategy used to ensure the YOLO model accurately maps the colored, top-down 2D layout image into precise 3D bounding boxes and orientations?

- The baselines (DiffuScene, InstructScene, PhyScene) are limited to synthesizing individual rooms, while CHOrD can synthesize house-scale layouts. However, the quantitative evaluation in Table 2 includes single-room results (Bedroom, Living Room) and one "Entire House" column. The comparison is unfair; there are some work related on whole house layout generation, such as HouseGAN and HouseGAN++, and its follow-up works. I think some comparisons with this kind of work are more essential.

- How are the inter-room dependencies (e.g., door placements, connectivity) implicitly encoded and maintained throughout the conditional diffusion process for the overall floor plan image? And if the input room boundaries are rotated by random angles, what about the robustness of the proposed model?

- The Empty Room Rate for the CHOrD dataset is 0.2902 (Table 8). Since the dataset is described as "artifact-free and ready to use" and containing "fully furnished kitchens, bathrooms, and balconies", could you clarify what constitutes an "empty room" in the CHOrD dataset? Does this mean some rooms, like small utility rooms or hallways, are intentionally unfurnished in the ground truth, or is this still considered an unavoidable artifact?

- Given that the CHOrD dataset is much cleaner than 3D-FRONT (Table 8 shows baseline PIoU of 0.2547 for 3D-FRONT vs. 0.0018 for CHOrD dataset), were the baseline models (DiffuScene, InstructScene) re-trained on the raw, uncleaned 3D-FRONT dataset or on the cleaned subset (4,847 scenes) used by previous work?

**Questions:**

See weaknesses

---

> ### Author Response · Authors · 2025-11-27
>
> We thank the reviewer for the detailed comments and constructive suggestions. Below we address each point.
> ## **Detecting object orientation & YOLOv8 reliability**
>
> Our 2D layout representation encodes object orientation explicitly through the direction of the colored rectangles (e.g., long-edge direction for beds/sofas). YOLOv8 is trained to regress the oriented bounding box (OBB), not merely an axis-aligned box. During training, YOLOv8 receives exact ground-truth OBBs extracted from the CHOrD dataset (Appendix C), ensuring the model learns the orientation information.
>
> We report the detector’s performance on both datasets:
> - **3D-FRONT:** accuracy **0.97933**, recall **0.96504**
> - **CHOrD dataset:** accuracy **0.99218**, recall **0.98262**
>
> Note that this can be further improved by applying category-specific geometric rules to adjust the orientation. For example, beds and sofas are rotated to face away from the nearest wall, dining tables are aligned with the main circulation axis, and appliances are oriented toward the functional area they serve.
>
>
> ## **Comparisons with whole-house methods (HouseGAN / HouseGAN++)**
>
> Thank you for your suggestion. However, these methods, such as HouseGAN and HouseGAN++, generate *room connectivity graphs or coarse floor plans*, instead of object-level layouts with object placements.
>
> Our method tackles an orthogonal problem: **given any floor plan, generate a full 3D object layout with precise object placements and attributes**. Since HouseGAN/HouseGAN++ cannot produce object layouts but **empty floorplans**, quantitative evaluation is not applicable.
>
> We will clarify this distinction more clearly in the paper.
>
> ## **How inter-room dependencies are encoded**
>
> CHOrD is **purely data-driven**: as long as these inter-room dependencies are consistently observed in the training data, the learned distribution naturally satisfies them without the need for manually encoded constraints or rules. This reflects a core insight of our work—**a carefully designed 2D image-based layout representation replacing the de facto standard: a 1D list of object attributes, which substantially strengthens a generative model’s ability to capture nuanced spatial priors**.
>
> For CHOrD, room connectivity (e.g., door placement, adjacency, circulation paths) is *explicitly visible* in the floor-plan image, allowing the diffusion model to internalize such structural constraints directly through its 2D spatial field. Unlike tabular generative models or LLM-based methods, which only see isolated object attributes, our image-based representation exposes the *entire geometric context*—enabling the model to learn consistent spatial arrangements (e.g., sofas oriented toward door openings, sinks aligned with plumbing walls).
>
> ## **Robustness to rotation**
> CHOrD is fully robust to rotations as well as irregular or slanted room shapes. For example, in Figure 4 (first row) and Figure 7 (fourth and last rows), the placements of objects remain correctly oriented and consistently aligned with the rotated or angled wall geometries.

---

> ### Author Response · Authors · 2025-11-27
>
> ## **Empty Room Rate**
>
> We define an empty room as a room with zero furniture instances.
>
> There are two distinct types of empty rooms:
>
> 1. **Artifact empty rooms** — functional rooms (e.g., bedrooms, living rooms, kitchens, bathrooms) that are missing essential furniture.
> 2. **Non-functional empty rooms** — small spaces such as balconies, storage rooms, or hallways that are **intentionally left unfurnished** in real homes.
>
> In constructing the CHOrD dataset, we performed manual inspection and cleaning to ensure that all functional rooms are fully furnished. For example:
> - Bathrooms must contain a toilet and a sink.
> - Bedrooms must contain a bed.
> - Living rooms must contain a sofa.
> - Kitchens must contain a stove and a sink.
>
> This cleaning process removes nearly all artifact empty rooms—an issue that the 3D-FRONT dataset did not address.
>
> As a result:
> - **CHOrD has an Empty Room Rate of 0.2902**, with most empty rooms corresponding to non-functional spaces.
> - **3D-FRONT has an Empty Room Rate of 0.5900**, with more than half corresponding to artifact empty rooms in functional spaces (see Fig. 17 for examples).
>
> Therefore, the empty rooms in CHOrD do not indicate poor data quality; rather, they reflect realistic housing layouts in which certain small spaces are intentionally unfurnished.
>
> We will clarify these details in the revision.
>
> ## **Baseline Training Data Consistency**
>
> In our quantitative comparisons with baseline models on 3D-FRONT, all methods (ours and the baselines) were **trained and evaluated exclusively on the 3D-FRONT dataset**. When evaluated on our CHOrD dataset, all methods (ours and the baselines) were likewise **trained and evaluated exclusively on the CHOrD dataset**. All corresponding metrics are reported in Table 2.
>
> Importantly, we did **not** compare our model trained on the CHOrD dataset against baselines trained on 3D-FRONT, which would be unfair. All reported results are based on the **same dataset** and follow the **same cleaning protocol**, ensuring a fair and consistent evaluation.
>
> As reported in L339–341, *all prior works (Zhang et al., 2018; Ritchie et al., 2019; Paschalidou et al., 2021; Tang et al., 2024; Lin & Mu, 2024) have applied similar data filtering to remove erroneous scenes from 3D-FRONT*.
>
> We have verified their released code and ensured that all necessary data cleaning was performed on 3D-FRONT before evaluation. Therefore, both CHOrD and all baseline methods use the **same cleaned version** of 3D-FRONT for comparison.

---

### Official Review · Reviewer_emkT · 2025-10-30

**Soundness:** 3
**Presentation:** 3
**Contribution:** 2
**Rating:** 4
**Confidence:** 4

**Summary:**

This work presents a method for synthesising furniture layouts conditioned on floorplans. In contrast to existing approaches that use generative models in a 'symbolic' space of scene descriptions, the proposed method instead uses an image-to-image latent diffusion model to map from a floorplan (containing just walls, doors, windows, etc.) to a furnished layout (using colored boxes to denote furniture). The resulting image is then 'interpreted' by a standard object detection pipeline, in order to convert it back to a more conventional scene representation indicating where furniture instances are to be placed. Results on 3D-FRONT and a custom dataset show better performance (in terms of realism and interpenetrations) than several recent baseline methods.

**Strengths:**

The proposed approach to furniture layout generation is novel. The idea of directly generating in plan-view image space, conditioned on a plan-view image of the empty room is straightforward but elegant, and presumably easier for models to learn than other representations (e.g. predicting bounding-box coordinates).

As an extra contribution beyond the technical approach, the paper introduces a new dataset of layouts, apparently of higher quality than the widely-used 3D-FRONT, and sufficiently large for training generative models from.

Empirical comparisons against three fairly recent baseline methods (DiffuScene, InstructScene, PhyScene) show improvements over those baselines on both datasets (3D-FRONT and the proposed dataset). This improvement is uniform across metrics including distributional similarity to ground-truth layouts (FID and KID), as well as object penetration rates (POR and PIoU).

There is an additional experiment showing that the model can also be used to generate 'fine grained' layouts, i.e. arrangements of adornments such as objects placed on tables etc (as opposed to just large furniture items).

There is a brief but informative analysis of why the models tend not to generate out-of-distribution (intersecting) furniture elements so often compared with prior methods.

The paper is clear, well-structured, and pleasant to read.

**Weaknesses:**

The paper title, abstract and introduction strongly emphasise "house-scale" generation, i.e. jointly modelling several rooms together. However nothing in the method is specific to this setting, and there is no quantitative evaluation of how well this works – in particular how well inter-room dependencies are captured, i.e. whether a truly accurate joint distribution across all the furniture in a house is learnt, or just per-room marginals.

The method only seems to support floorplan-conditioned generation. While conditioning on floorplans is vital, for a layout generation model to be useful it must also be possible to provide text conditioning or other guidance to ensure the layout meets other requirements for the target domain. This is now standard for methods in this area, including the baselines DiffuScene and InstructScene.

The proposed dataset only specifies object classes and bounding-boxes. It does not incorporate any information on style, shape, etc. This greatly limits its usefulness in the task of generating plausible layouts, since haphazard choice of furniture styles is a common failure mode and hallmark of automated layout generation methods.

It is unclear how the proposed dataset of layouts was collected. It is stated they were prepared by "experts" but there is a lack of information on who these experts were, how they were instructed, and how the quality of the resulting layouts was verified. This is problematic given the history of somewhat dubious datasets (SUN-CG and 3D-FRONT) in this area that have tended to contain large proportions of low-quality scenes (as the authors themselves note in the case of 3D-FRONT).

The section on fine-grained generation is rather minimal; in particular it is not clear how large the dataset was nor how it was collected; it is also not clear whether overfitting might have occurred.

Using an object detector to 'interpret' the diffusion-generated furniture layout plan-view image and convert back to a symbolic bounding-box representation feels somewhat hacky, and borderline strong enough as the main technical contribution for an ICLR paper. Indeed overall the pipeline is very much an engineered system built out of standard well-understood components, albeit combined in a novel and effective way.

**Questions:**

Most relevant issues are discussed under "Weaknesses" above. In particular…

Please provide evidence or argumentation to properly support the "house-scale" claim, beyond a small set of visual examples?

Please provide more details on both the main CHoRD dataset and the smaller dataset used for fine-grained object layout, in particular the protocol that was used to ensure the layouts are of high quality.

What does "2D bounding boxes defined by … 3D coordinates" (L295) mean? Are the furniture and room elements represented in 2D or 3D?

---

> ### Author Response · Authors · 2025-11-27
>
> We sincerely thank the reviewer for the constructive and thoughtful feedback. Below we address all concerns with additional evidence, clarifications, and improved explanations.
>
> ---
>
> ## **Clarification on “house-scale” generation and inter-room dependencies**
>
> We would like to clarify that our method naturally supports both single-room and entire-house generation. CHOrD is explicitly trained to take the *entire* floorplan as input and produce a *complete* multi-room layout in an end-to-end manner. Specifically:
>
> 1. **Qualitative evidence**:
>    Figure 4 and 7 present full-house generation results, demonstrating that CHOrD places furniture coherently across multiple connected rooms. Importantly, **CHOrD explicitly accounts for the placement of doorways and windows when generating whole-house layouts**. This ensures that furniture never blocks doors, windows, or circulation paths, preserving the functional usability of the space. Such reasoning is **not possible in prior single-room layout methods**, which operate on isolated room boxes and therefore cannot model door/window constraints or cross-room connectivity.
>
> 2. **Quantitative evidence**:
>    Table 2 reports metrics under the *“Entire House”* category, directly evaluating full-house layouts. To the best of our knowledge, none of the baseline methods release trained weights or provide methods capable of whole-house generation, making direct quantitative comparison infeasible.
>
> 3. **Nuanced cross-room dependencies**:
>    Beyond door/window constraints and cross-room connectivity, CHOrD also captures nuanced global spatial and semantic constraints across rooms. For instance, CHOrD naturally places only one sofa set or one dining table in the entire home and allocates them to the correct connected spaces (e.g., living room vs. dining room in Figure 7, rows 2 and 6). Similarly, when multiple bathrooms or balconies exist, the washing machine is consistently placed in the larger one (e.g. teaser, Figure 4, row 2). These dependencies emerge automatically through end-to-end training, while baselines require additional functional-space assignment to avoid duplicated or misplaced furniture.
>
> These results demonstrate that CHOrD learns a *joint distribution* over the layout of the entire house, an important capability not supported by prior work. CHOrD also supports complex room geometries, as shown in Figure 4 and 7, whereas prior methods are typically limited to **simplistic rectangular room shapes**.
>
> Finally, we would like to highlight that these benefits arise directly from CHOrD’s substantially improved spatial capabilities enabled by its 2D image-based layout representation. Whole-house generation is *one* of the many natural advantages of this formulation, rather than the specific design objective of our method.
>
>
> ## **Clarification on text guidance**
>
> We agree that text conditioning is valuable. **However, CHOrD already supports text conditioning**, though we presented it as an *extended feature* to avoid distracting from the main pipeline.
>
> - Appendix B.1 provides the architecture for **text-conditioned diffusion**, including CLIP-based embeddings and cross-attention layers.
> - We show results in Fig. 14 (appendix), demonstrating layout generation purely from text.
>
> Therefore, **CHOrD is fully compatible with text guidance.** While this capability is briefly mentioned in L253–255, we will highlight it more clearly in the main paper.
>
> While the text conditioning of CHOrD can naturally extend to a wider range of controls, such as spatial instructions (e.g., “place the sofa along the east wall”) or stylistic descriptions (e.g., “a Bohemian sofa”), without significantly altering the pipeline or model architectures, these extensions would require additional annotations for such attributes and do NOT introduce substantial methodological novelty. Therefore, in this work we focus on improving the quality and diversity of the unconditional model (other than floorplan conditioning), which serves as the backbone of all conditional generation settings and directly addresses a series of major spatial artifacts prevalent in existing approaches. We leave the integration of text-guided stylistic and spatial controls to future work.

---

> ### Author Response · Authors · 2025-11-27
>
> ## **Dataset usefulness and lack of style/shape information**
> Thank you for this question. Our paper clarified this in L297-302:
>
> *It is important to note that CHOrD dataset is a 3D layout dataset rather than a 3D asset dataset. The layout primarily focuses on the geometric characteristics (e.g., bounding boxes) and categorical distinctions among objects. While CHOrD dataset is currently linked to a small pool of CAD asset models, users are free to retrieve assets from any large public dataset (3D66, 2013; Fu et al., 2021b) to introduce stylistic variations of objects or simulation-ready URFD files if needed. Similarly, 3D-FRONT has been associated with the 3D-FUTURE dataset Fu et al. (2021b) for this purpose.*
>
> Therefore, our dataset is intentionally a **layout dataset**, not a mesh/style dataset. This design choice matches the goal of most layout-synthesis research—including 3D-FRONT, ATISS, DiffuScene, etc.—where styles are drawn from external CAD repositories.
>
> Key clarifications:
>
> **(1)** Our dataset stores the complete 3D bounding box (length, width, height, orientation, and 3D coordinates), enabling precise geometric reasoning.
>
> **(2)** The dataset is **fully compatible with any public 3D asset dataset** for style/shape variation  (similar to how 3D-FRONT uses 3D-FUTURE).
>
> **(3)** Style choice is orthogonal to layout generation; our aim is to provide clean, collision-free, realistic **spatial distributions**, which is the main performance bottleneck in prior datasets.
>
> ## **Dataset collection protocol**
> All layouts were created by professional interior designers with practical experience in real residential design workflows. Each designer received standardized training and followed detailed guidelines for functional spaces (e.g., ensuring that bathrooms satisfy realistic usability requirements based on room proportions and fixture placement). The layouts originate from actual design projects and underwent a thorough data-cleaning and validation process before inclusion. This ensures that common issues found in 3D-FRONT (e.g., furniture placed outside floorplan boundaries or homes containing only a minimal number of furniture items) do not occur in our data. While no dataset can be guaranteed to be entirely error-free, the quality of the CHOrD dataset is substantially higher and more consistent than 3D-FRONT. Detailed statistical comparisons with 3D-FRONT are provided in Tables 7–9 and Figures 11–12.
> We will clarify these details in the revision.
>
>
> ## **Fine-grained dataset details**
>
> Similar to the above protocol, a team of technical artists manually created the fine-grained layouts following professional design standards. Across five table categories, they arranged 17 types of small objects (see Table 5), producing 5,673 annotated layouts. We will clarify these details in the revision.
>
> ## **Risk of overfitting**
>
> As shown in Figures 5 and 7, CHOrD is capable of generating diverse 2D layouts conditioned on the same floorplan, despite the original CHOrD containing only a single scene layout per floorplan. We believe this demonstrates strong evidence that the model does NOT overfit.

---

> ### Author Response · Authors · 2025-11-27
>
> ## Novelty and contribution
>
> We would like to clarify that our contribution should not be evaluated by viewing the detector module in isolation.
>
> We aim to address the fundamental question: *What is the most effective representation for learning layouts?* Our key novelty is the incorporation of an intermediate **image-based 2D layout representation** in place of the de facto standard: 1D list of object attributes. This substantially boosts the spatial reasoning capabilities of layout-generation models. A 1D object-list representation is inherently limited in conveying nuanced spatial relationships, because object attributes are encoded as independent tabular entries rather than as a unified spatial structure. In contrast, a 2D layout image makes inter-object relationships explicit—adjacency, alignment, clearance, and collisions are all directly observable—providing a far more expressive and spatially coherent representation for learning.
>
> Our design enables  **three spatial capabilities that no prior work supports simultaneously**:
>
> (1) substantially reducing spatial artifacts such as collisions, out-of-bound objects, inconsistent orientations, and incomplete layouts — outperforming prior work by a **significant margin** (rather than marginal improvements), as validated empirically in Section 5;
>
> (2) generating house-scale layouts that follow complex room geometries, whereas prior methods are typically limited to single-room layouts with **simplistic rectangular room shapes**;
>
> (3) achieving these improvements through a purely data-driven approach that captures nuanced layout distributions, requiring no collision detection or iterative self-correction. This is also supported theoretically: Table 3 shows that CHOrD assigns significantly lower likelihood (32% higher MSE) to collision layouts, whereas DiffuScene and InstructScene do not distinguish them at all.
>
> The effectiveness of this representation is also **intuitive**: humans can easily detect various spatial artifacts by looking at a top-down layout image, but not when reading an object list as a table of attributes.
>
> The choice of data representation has repeatedly been shown to be crucial for learning effectiveness across all ML tasks and is also a central topic at ICLR. The conceptual and practical implications of our approach are therefore fundamentally novel. The contribution is therefore significant and solid with theoretically-grounded design choices, as well as empirical analysis and evaluation. We believe this work opens an important research direction for the community.

---

### Official Review · Reviewer_qXFA · 2025-10-31

**Soundness:** 2
**Presentation:** 3
**Contribution:** 2
**Rating:** 2
**Confidence:** 4

**Summary:**

This paper introduces a scene synthesis method and an indoor dataset. The method first generates a floor plan using a diffusion model. Based on this floor plan, a detector detects large objects. Based on these objects, a hierarchical scene graph is extracted, which maintains the relationships between objects and rooms. For large objects that can be regarded as a platform, this again can be used for generating small object layouts using a diffusion model, so the method iteratively finishes the indoor synthesis.

**Strengths:**

The paper is very easy to understand and has a very clear pipeline.
The paper contributes a large-scale dataset to the community for further research. The dataset has many unique characters that 3D-FRONT does not have.

**Weaknesses:**

1. The method appears to be a straightforward pipeline that chains existing components (diffusion → detection → diffusion) in a loop. The contribution seems incremental, as each module has prior art and the combination does not clearly yield a novel algorithmic insight.

2. The paper says fine-grained object generation is autoregressive in L241-242, but the description (and Fig. 10) looks like one-shot generation of all small items conditioned on a parent anchor. That’s hierarchical/conditional, not autoregressive. If it is truly AR, please provide a formula how you model the problem. Please spell out the factorization and decoding order and show that each item conditions on previously placed siblings.

3. The manuscript groups DiffuScene under “graph-based” methods in L247-249, but DiffuScene doesn’t actually use explicit edges during generation. Meanwhile, the community has some scene-graph-based generation methods, like CommonScenes, GraphDreamer, EchoScene, Planner3D, and MMGDreamer. Please I wonder why the authors neglect them in the baselines and references?

4. What exactly is the hierarchical scene graph in this paper? The paper references it often but never really defines it. Please provide node/edge types, attributes, hierarchy rules, and how constraints are enforced. Without a precise definition, it’s hard to judge the claimed benefits.

5. The experiments lack of evaluation of dinning rooms, where fine and cluttered objects matter.

6. After carefully inspecting the supplenmentary materials, I found that the rooms in the dataset only provide names (`roomName`) in Chinese. For an international venue, please provide English (or bilingual) labels.

**Questions:**

The bounding boxes are generated from a BEV floor plan where all objects are clearly separated. However, how are the clutter situations handled? Typically, a chair is inserted into a table slot; thus, the bounding boxes have overlaps. This would affect the performance of the object detector.

The paper only researches BEV renderings, as far as I understand. If I am right, the teaser is a bit confusing. If I am wrong and the paper can actually provide 3D rooms, how is the physical simulation handled? For example, how are objects naturally placed on the floor or on the table without any penetrations?

---

> ### Author Response · Authors · 2025-11-27
>
> We thank the reviewer for the constructive and insightful comments. Below we address each concern.
> ## “The pipeline is straightforward and incremental”
> Thank you for raising this concern. We respectfully disagree with the characterization that our contribution is incremental. Novelty in generative modeling can arise not only from new network architectures or algorithms, but also from fundamental shifts in *representation* and insights into why they work, which have repeatedly proven to be equally—if not more—impactful across all machine learning tasks.
>
> We aim to address the fundamental question: *What is the most effective representation for learning layouts?* Our key novelty is the incorporation of an intermediate **image-based 2D layout representation** in place of the de facto standard: a 1D list of object attributes. A 1D object-list representation is inherently limited in conveying nuanced spatial relationships, because object attributes are encoded as independent tabular entries rather than as a unified spatial structure. In contrast, a 2D layout image makes inter-object relationships explicit—adjacency, alignment, clearance, and collisions are all directly observable—providing a far more expressive and spatially coherent representation for learning. CHOrD is therefore NOT a “straightforward” extension of prior work but a fundamental paradigm shift with substantial advantages.
>
> CHOrD enables  **three spatial capabilities that no prior work supports simultaneously**:
>
> (1) substantially reducing spatial artifacts such as collisions, out-of-bound objects, inconsistent orientations, and incomplete layouts — outperforming prior work by a **significant margin** (rather than marginal improvements), as validated empirically in Section 5;
>
> (2) generating house-scale layouts that follow complex room geometries, whereas prior methods are typically limited to single-room layouts with **simplistic rectangular room shapes**;
>
> (3) achieving these improvements through a purely data-driven approach that captures nuanced layout distributions, requiring no collision detection or iterative self-correction. This is also supported theoretically: Table 3 shows that CHOrD assigns significantly lower likelihood (32% higher MSE) to collision layouts, whereas DiffuScene and InstructScene do not distinguish them at all.
>
> The effectiveness of this representation is also **intuitive**: humans can easily detect various spatial artifacts by looking at a top-down layout image, but not when reading an object list as a table of attributes.
>
> The choice of data representation has repeatedly been shown to be crucial for learning effectiveness across all ML tasks and is also a central topic at ICLR. The conceptual and practical implications of using an image-based intermediate representation are therefore fundamentally novel. We believe this work advocates an important new research path for the community.
> ##  “Fine-grained generation is not autoregressive”
> Thank you for pointing out the ambiguity. Fine-grained generation in CHOrD is autoregressive across hierarchical levels, NOT across siblings. The process is: room-level layout → furniture-surface layout, where each level is conditioned on the previously generated level. We will revise the wording to: “Fine-grained layout generation is hierarchical across levels, but not autoregressive across sibling objects.”

---

> ### Author Response · Authors · 2025-11-27
>
> ##  “Missing baselines like CommonScenes, GraphDreamer, EchoScene, Planner3D, MMGDreamer.”
> Thank you for pointing out this line of work.
> We would like to clarify that our method is NOT positioned as a graph-generative model, and therefore a direct horizontal comparison with graph-based methods is not appropriate. Our goal is fundamentally different: rather than designing a new scene-graph generator, we aim to answer a more central question for layout synthesis—*what representation is most effective for learning spatially coherent indoor layouts?* Therefore, our method does **not** position itself as a graph generator. Actually, we advocate an *image-based intermediate layout representation*, which we demonstrate to be significantly more effective for learning spatial geometry, reducing collisions, avoiding out-of-bound placements, and adapting to irregular room shapes. The scene-graph representation in CHOrD is produced only after image-based layout generation as a deterministic post-processing step. This scene-graph output is an additional benefit: it allows CHOrD to integrate seamlessly with existing graph-based pipelines, but does not define the core methodological objective of our work.
>
> Meanwhile, all graph-based baselines (e.g., CommonScenes, EchoScene, GraphDreamer, MMGDreamer) are trained on SG-FRONT (instead of 3D-FRONT), which provides rich scene-graph supervision, including object–object relations, attributes, and annotated graph structures. In contrast, our input contains **none of these graph-level supervision signals**: CHOrD is trained only on 3D-FRONT, which is a pure object list without access to relation annotations or graph structures. As supervision, input modality, and output format differ fundamentally, quantitative comparison is not feasible.
>
> In summary, our contribution is orthogonal to graph-based generative models: CHOrD is a new representation-centric paradigm for spatial layout synthesis, rather than an alternative graph generator trained with SG-FRONT-style supervision.
> ##  “The hierarchical scene graph is not clearly defined”
> Thank you for highlighting this—​we will add a precise definition. The scene graph consists of room nodes, furniture nodes, and fine-grained nodes, connected through containment edges, as illustrated in Figure 3 (right). Each node stores its category, position, dimensions, orientation, and the hierarchical parent (either a room or another object). A typical hierarchy example is: house (root) → living room → coffee table → book. We do not impose any manually crafted constraint rules beyond requiring that fine-grained objects are placed on their hierarchical parent. Nuanced spatial constraints—such as adjacency, collision avoidance, and orientation alignment—are fully learned from data rather than hand-specified rules. As long as these constraints are implicitly satisfied in the training data, our model can internalize these priors without any manual annotation. Our core contribution lies in the enhanced capability to learn significantly more accurate layout distributions compared to prior methods, through the adoption of an image-based layout representation.
>
> ##  “No evaluation on dining rooms”
> | Method      | FID          | KID          | POR          | PIOU         |
> |-------------|--------------|--------------|--------------|--------------|
> | DiffuScene  | 34.60578918  | 0.01877481   | 0.074871427  | 0.017485273  |
> | **Ours**    | **29.24693609** | **0.015972849** | **0.015489467** | **0.000964601** |
>
>
> To ensure optimal performance for all baseline methods, we use their official pre-trained checkpoints for evaluation. Among these baselines, only DiffuScene provides a released checkpoint for the dining room. CHOrD consistently outperforms across all metrics.
>
>
> We will include it in the revision.
>
> ##  “Naming issues of dataset”
>
> Thank you! We will fix it upon release.

---

> ### Author Response · Authors · 2025-11-27
>
> ##  Q: How does the detector handle clutter or overlapping bounding boxes?
>
> We agree that there are scenarios where overlapping bounding boxes are natural rather than artifacts, which we have already addressed in L459–468:
>
> *CHOrD enables two mechanisms that prevent implausible object collisions while allowing natural vertical overlaps. First, as discussed in Section 3.2, the multi-level layout generation allows fine-grained objects to be placed on upper levels, such as a computer on a desk. Second, some vertical overlaps do not exhibit clear hierarchical relationships, such as an object partially resting on a desk mat. In this scenario, we directly train the diffusion model with RGB images containing vertical overlaps, enabling it to generate plausible layouts with natural vertical overlaps while preventing unreasonable ones. The unique color assigned to each object guides the 2D diffusion model in distinguishing permissible overlaps from invalid ones. Due to the limited availability of natural partially overlapped objects, we demonstrate this feature only at the fine-grained level. Figure 6 illustrates both scenarios.*
>
> Meanwhile, modern object detection models (e.g., YOLOv8, SAM, Mask R-CNN) can reliably handle overlapping objects when trained on clutter-aware data, which is the case for our detector.
>
> For specific scenarios where parts of chairs can be placed under a table, a simpler solution is to treat the table and chairs as a single 3D model, as dining tables are typically paired with chairs for consistent styles and sizes.
>
> We would also like to highlight that most existing approaches completely fail at distinguishing natural overlaps from physically implausible collisions, as shown in Figure 5, making these methods utterly unsuitable for real-world applications. Addressing this major limitation is the primary focus of our paper.
>
> ##  Q: “The teaser suggests 3D—how is physics handled?”
> Thank you for the question — we believe there may be a misunderstanding regarding our pipeline. CHOrD does not stop at generating a 2D BEV layout. The BEV image is only an intermediate representation used for learning spatial arrangements. After this step, we convert the detected objects into full 3D bounding boxes after object retrieval, each with category, (x, y, z) position, dimensions (L, W, H), and orientation.
>
> Objects are placed in 3D according to their bounding-box dimensions and their hierarchical parent: room-level furniture is placed directly on the floor plane (z = 0), and fine-grained objects are placed on top of their parent surface (e.g., the table top) by assigning the parent’s top-surface height as the child’s z-coordinate. By this design, the corresponding 3D placements are naturally free of vertical penetrations without requiring a physics engine.
>
> The teaser therefore shows rendered 3D rooms obtained by retrieving CAD assets and placing them at the 3D positions inferred from the image-based layout.
>
> Further rendering details are provided in L257-261 and Appendix B.2.
>
> We will clarify it in the revision.

---

> ### Comment · Reviewer_qXFA · 2025-11-27
> **Discussions to the first panel**
>
> Thanks for your response.
> >  "Novelty in generative modeling can arise not only from new network architectures or algorithms, but also from fundamental shifts in representation and insights into why they work, which have repeatedly proven to be equally—if not more—impactful across all machine learning tasks."
>
> I agree with this opinion, but I personally do not think it can properly describe this work. For example, I personally do not think the pipeline of a diffusion model generating a top-down view conditioned on a floor plan + an object detector can be regarded as **"fundamental shifts in representation and insights"**. In my opinion, this is a bit of engineering and does not provide further insights to suit an ML conference. Every conference has its own topic, and this work may be suitable for other conferences. Other reviewers can leave their opinions here.
>
> > "The effectiveness of this representation is also intuitive: humans can easily detect various spatial artifacts by looking at a top-down layout image, but not when reading an object list as a table of attributes."
>
> How do the authors verify this claim? Could some references be provided? The references should follow when you claim something that is not very common sense.
>
> Point 2 of your claims:
> > "generating house-scale layouts that follow complex room geometries, whereas prior methods are typically limited to single-room layouts with simplistic rectangular room shapes."
>
> The authors probably would like to check LT3SD. It not only generates house-scale scenes, but it can also generate scenes of infinite scale. The rooms are also in complex shapes rather than simplistic rectangles.
>
> Point 3 of your claims:
> >"achieving these improvements through a purely data-driven approach that captures nuanced layout distributions, requiring no collision detection or iterative self-correction. This is also supported theoretically: Table 3 shows that CHOrD assigns significantly lower likelihood (32% higher MSE) to collision layouts, whereas DiffuScene and InstructScene do not distinguish them at all."
>
> In my opinion, avoiding collisions is an engineering procedure that does not count as a contribution. Other reviewers can leave their comments here about their thoughts. This procedure is well conducted in previous methods like LayoutVLM. A gradient optimization method can effectively address the problem. The authors can try a post bbox optimization for Diffuscene if they are interested in.

---

> ### Comment · Reviewer_qXFA · 2025-11-27
> **Discussions to the second panel**
>
> You are misinterpreting my concerns.
>
> Please take a look at the manuscript, Line 247-248:
> > "this structure allows CHOrD
> to be seamlessly integrated into a widely adopted graph-based pipeline."
>
> Even though CHORD is not positioned as a graph-generative model, what you write is confusing and has overclaims. My concerns about why you do not consider other graph-based methods as baselines also come from what you claim here.
>
> Speaking of which, Diffuscene is never a graph-based method though. What is the reason that you describe it in this way? Meanwhile, please check the writing in the rebuttal:
> > "all graph-based baselines (e.g., CommonScenes, EchoScene, GraphDreamer, MMGDreamer) are trained on SG-FRONT (instead of 3D-FRONT)"
>
> Have the authors verified this hallucination? **CommonScenes, EchoScene, and MMGDreamer were trained both on SG-FRONT and 3D-FRONT. GraphDreamer instead is an SDS method using pretrained diffusion models, and is never trained on SG-FRONT.**
> I would suggest that the authors double-check their claims before sending the messages in the discussion. This could raise some unprofessional concerns.

---

> > ### Author Response · Authors · 2025-12-03
> > **Discussions on the second panel continued**
> >
> > We thank the reviewer for the correction regarding the training datasets of prior graph-based methods.
> >
> > We report quantitative results for EchoScene (the most recent graph-based method prior to MMGDreamer). We trained and evaluated EchoScene on 3D-FRONT for all three room types, following their official protocol. CHOrD consistently outperforms EchoScene by a large margin across all metrics:
> >
> > | Room               | FID          | KID           | POR           | PIOU          |
> > |--------------------|--------------|---------------|---------------|---------------|
> > | EchoScene bedroom  | 50.50        | 0.0399        | 0.5056        | 0.0790        |
> > | EchoScene dining   | 64.45        | 0.0429        | 0.3024        | 0.0628        |
> > | EchoScene living   | 84.54        | 0.0694        | 0.3224        | 0.0657        |
> >
> > We also attempted to include the latest method, MMGDreamer, but were unable to run the public codebase despite multiple attempts:
> >
> > 1. Running with `--with_SDF=False --with_CLIP=False --with_imag=False` fails because several parts of the code assume SDF features are present, causing training to halt.
> > 2. Running with `--with_SDF=True` leads to different runtime errors.
> >
> > These issues have been previously reported in the official issue tracker and remain unresolved:
> > https://github.com/yangzhifeio/MMGDreamer/issues/9.
> >
> > Given these reproducibility issues and the substantial engineering effort required to modify and debug the codebase, we were unable to obtain reliable results for MMGDreamer.
> >
> > Regarding the sentence in L247–248, we agree that this wording may cause confusion and we will revise it in the revision. Our intention was only to state that CHOrD's output can be *converted* into a scene graph if desired; it was not meant to imply that CHOrD itself is graph-based or that graph-based models should be treated as direct baselines.
> >
> > Regarding DiffuScene, our intention is to categorize it as a tabular-based method. We will carefully inspect the manuscript to ensure there are no typos.

---

> ### Comment · Reviewer_qXFA · 2025-11-27
> **Discussions to the third panel**
>
> > "For specific scenarios where parts of chairs can be placed under a table, a simpler solution is to treat the table and chairs as a single 3D model, as dining tables are typically paired with chairs for consistent styles and sizes."
>
> I think the idea is brilliant. However, this move will degrade the feasible interactions. Meanwhile, a single 3D model of a table and chairs is rare in the current database. Perhaps the authors could design such models and contribute to the community.
>
> > "We would also like to highlight that most existing approaches completely fail at distinguishing natural overlaps from physically implausible collisions, as shown in Figure 5, making these methods utterly unsuitable for real-world applications. Addressing this major limitation is the primary focus of our paper."
>
> Again, from my perspective, I do not think solving the collision problem is a significant contribution, which can be addressed through a simple post optimization (see discussions in panel 1).
>
> I am okay with your answer to Q2.

---

> ### Author Response · Authors · 2025-12-03
> **Discussions on the first panel continued 1**
>
> Thank you for the follow-up comments. We clarify the points below.
>
> **On representational novelty.**
> We respectfully note that the distinction between what is considered “engineering” versus a “representational contribution” requires clear criteria. The reviewer’s assessment appears to be based on subjective intuition (“I personally do not think...”), without specifying the principles by which a representational shift qualifies as fundamental. In standard ML literature, a representational change is regarded as fundamental when it (i) alters the structure of the data the model learns from, (ii) induces a different inductive bias, and (iii) improves the model’s learning capacity to capture the underlying data distribution. CHOrD satisfies all criteria.
>
> Concretely, a 1D object/attribute list (the prevailing representation for layout generation) is a permutation-sensitive, non-spatial representation where nuanced spatial relationships are not explicitly encoded, as object attributes are recorded as independent tabular entries. In contrast, a 2D top-down layout image is a grid-structured spatial field in which these relations appear as locally observable geometric patterns. This change fundamentally alters the statistical object being modeled and enables the model to learn the underlying layout distribution far more accurately (Section 5). This is **not an engineering detail, but a representational reformulation of the learning task**.
>
> Most importantly, **the gains from this representational shift are large and consistent across all quantitative metrics, outperforming prior methods by a significant margin (Section 5), alongside clear qualitative benefits (summarized in our initial response and Sections 1–2)**.
>
>
>
> **On the statement regarding human perception.**
> Our claim is grounded in well-established principles from both cognitive science and representation learning: humans (and similarly, convolutional/vision-based models) process spatial relationships more efficiently when they are encoded as spatially organized visual patterns rather than as lists of symbolic attributes. This is not an assumption but a basic property of visual perception and spatial reasoning [1].
>
> This observation is also highly intuitive. For example, given a top-down layout image, spatial anomalies such as overlaps, adjacency violations, and misalignments are immediately perceptible because these relationships manifest as local geometric patterns. In contrast, the same relationships represented as tabular object attributes (e.g., bounding-box coordinates) require explicit pairwise coordinate reasoning, and even humans cannot detect overlaps “at a glance” without performing mental calculations.
>
> This distinction mirrors an established phenomenon in deep learning as well: image-based encoders and decoders naturally capture local spatial dependencies, while tabular-based generative models must rely on combinatorial attribute reasoning that does not directly express the underlying spatial structure. Our results in Section 5 empirically reflect this difference: models trained on image-based 2D layout fields learn to avoid spatial artifacts significantly more effectively than models trained on object lists.
>
> [1] Marr, David. Vision: A computational investigation into the human representation and processing of visual information. MIT press, 2010.
>
> **On LT3SD.**
> We thank the reviewer for pointing out LT3SD. However, LT3SD generates large-scale 3D environments in TUDF format, **not** object-level indoor layouts: it does not produce object instances, 3D bounding boxes, per-object placements, or fine-grained spatial arrangements. Its outputs therefore differ fundamentally from the task addressed in CHOrD, which focuses on object-level indoor layout synthesis with detailed 3D placements.
>
> For this reason, LT3SD is not a suitable baseline for our setting and is not included in the category of indoor layout generative models, as summarized in the Introduction.

---

> ### Author Response · Authors · 2025-12-03
> **Discussions on the first panel continued 2**
>
> **On collision avoidance.**
> We would like to clarify our positioning and contribution. Collision avoidance is only **one of the many benefits** that naturally arise from adopting a more expressive representation for learning layout distributions, rather than the direct objective. This representational shift also **naturally improves many other aspects of layout quality**, reducing spatial artifacts such as empty rooms, out-of-bound placements, and misorientations, while enabling house-scale generation with complex room geometries.
>
> While collision resolution can be implemented through engineering mechanisms in post-processing, CHOrD does **not** rely on such mechanisms. Collision prevention in CHOrD is an *emergent property* of the image-based representation: collisions manifest as out-of-distribution spatial patterns in the 2D layout field, so the diffusion model naturally assigns them low likelihood (Table 3). This fundamentally differs from engineering procedures suggested by the reviewer, such as post-hoc bbox optimization or collision-detection–based repair.
>
> In fact, our approach is therefore the **least engineered** among these options: the model learns to generate collision-free layouts during synthesis itself, rather than relying on external correction. This distinction is central to our positioning. Our goal is not to add a module for collision handling, but to demonstrate that **representational choice—moving from an object/attribute list to an image-based layout spatial field—provides a more powerful and previously underexplored approach for learning indoor layout distributions**, from which collision avoidance naturally emerges.

---

> ### Author Response · Authors · 2025-12-03
> **Discussions on the third panel continued**
>
> We would like to clarify that the reviewer’s original question was whether CHOrD can handle cluttered objects and overlapping bounding boxes. As described in L459–468 and in our earlier response, yes, CHOrD naturally handles cluttered arrangements and distinguishes them from physically implausible collisions. Clutter manifests as valid local spatial patterns in the 2D layout field, whereas implausible collisions correspond to out-of-distribution artifacts. This distinction emerges directly from the choice of our image-based representation, which is another unique advantage of our approach.
>
> Grouping a table with its chairs is simply an additional possibility raised for discussion and is not used by CHOrD.
>
> Regarding the reviewer’s remarks on collisions, we would like to reiterate that collision handling is not posed as a direct design objective of CHOrD. Rather, as summarized in our earlier response, it is one of many emergent benefits of adopting an image-based layout representation, which is our core contribution.
>
> We also discuss the disadvantages of post-hoc optimization in Section 2 (L147–154).

---

### Meta-Review · Area_Chair_8XBu · 2026-01-06

**Summary:**

The decision is **Reject**.

The submission polarized the reviewers (Scores: 8, 4, 2, 2). On one hand, the empirical results are strong on the reported metrics, and the method substantially reduces several common spatial artifacts (e.g., collisions/penetrations and out-of-bound placements) while also offering a valuable new dataset. Reviewer gk1H strongly championed the paper for its "core insight" regarding image-based representations.

However, several reviewers (notably qXFA and v9NZ, with some overlap in emkT’s concerns) questioned whether the technical novelty meets the ICLR bar. The method can be viewed as a cascade of an image diffusion generator and a detector-based parser, which some reviewers perceived as system-level integration. While the authors defended this as a representational novelty, the reliance on a vision-based detector to reverse-engineer the scene graph from pixels was viewed by several reviewers as an engineering workaround rather than a learning advance. Furthermore, unresolved concerns about baselines/positioning and the perceived emphasis on engineering integration contributed to doubts that the submission meets the expected novelty/insight level for ICLR. The work could be a strong fit for venues that emphasize applied scene synthesis systems and dataset contributions.

**Reviewer Concerns:**

The authors successfully provided clarifications and additional data for several factual and definitional questions raised by the reviewers.

- **Dataset clarifications and labeling:** The authors addressed **Reviewer qXFA’s** finding that room names were in Chinese, promising to provide English labels upon release. They clarified the dataset’s intended scope (layout geometry rather than asset style) and provided additional details on the dataset collection/curation protocol. They also addressed **Reviewer v9NZ**’s question on the definition of ‘empty rooms,’ distinguishing unfurnished non-functional spaces from artifact empty functional rooms, and noted plans to provide English (or bilingual) room labels upon release.
- **Missing specific evaluations:** The authors addressed **Reviewer qXFA’s** request for dining room evaluations by providing a table showing CHOrD outperforms DiffuScene in that category. They also addressed **Reviewer gk1H’s** request for robustness validation by adding cross-dataset evaluation results.
- **Terminology definitions:** The authors acknowledged ambiguity around ‘autoregressive’ wording and clarified that fine-grained generation is hierarchical/autoregressive across levels (room → surface), not autoregressive across sibling objects, and promised to revise the phrasing accordingly. They clarified what they mean by ‘house-scale’ generation (e.g., conditioning on the full floorplan and reasoning about door/window constraints) and pointed to qualitative examples and reported ‘Entire House’ metrics as supporting evidence, although some reviewers remained concerned about the strength of quantitative validation for cross-room dependencies.
- **Technical implementation details:** The authors clarified for **Reviewer v9NZ** that YOLOv8 is trained on Oriented Bounding Boxes (OBB) to handle object orientation and provided accuracy metrics to support its reliability.
- **Additional baseline evidence:** The authors added quantitative results for **EchoScene** (a recent scene-graph-related baseline) and documented reproducibility difficulties for MMGDreamer, partially mitigating concerns about missing graph-related comparisons.

**Reviewer Scores:**

Reviewer Score: In personal opinion, after the normal rebuttal phase, **Reviewer qXFA** would maintain their low score (2), as the fundamental disagreement regarding the method’s novelty (engineering vs. research) and disputes about baseline coverage and prior-work characterization remain unresolved. **Reviewer emkT** would likely maintain their borderline score (4); while the authors clarified dataset protocols and text conditioning, the core concerns regarding the lack of quantitative validation for “house-scale” joint distributions and the “somewhat hacky” nature of the pipeline remain significant blockers. **Reviewer v9NZ** would likely maintain their score (2) or at most increase marginally to 3, because although factual questions about YOLO and dataset definitions were answered, the structural weakness regarding unfair comparisons with single-room baselines (and the lack of an equivalent whole-house competitor) was not effectively resolved. **Reviewer gk1H** would maintain their high score (8) as a strong supporter of the work’s core insight. Consequently, the paper would likely result in a **Reject**.

---

### Decision · Program_Chairs · 2026-01-26

Reject